

# Differences in key volatile organic compound species in ozone formation between their initial and measured concentrations

Xudong Zheng, Shaodong Xie*

5 College of Environmental Science and Engineering, State Key Joint Laboratory of Environmental Simulation and Pollution Control, Peking University, Beijing 100871, China

*Correspondence to*: Shaodong Xie (sdxie@pku.edu.cn).

**Abstract.** To reduce the uncertainties in identifying key volatile organic compounds (VOCs) species influencing ozone ($O_3$) formation based on observed VOCs concentrations, this study proposed key species identification from the initial VOCs concentrations. The initial VOCs concentrations during the daytime and nighttime were calculated using reaction rates and 10 hourly measured 99 VOCs concentrations at Deyang, Chengdu, and Meishan, southwest China during summer. The initial concentrations of alkenes and aromatics were higher than the measured ones. The largest differences between initial and measured concentrations were 1.04 ppbv for cis-2-butene at Deyang, 0.86 ppbv for isoprene at Chengdu, and 1.98 ppbv for isoprene at Meishan, respectively. Due to secondary production, the initial concentrations of oxygenated VOCs were lower than the measured ones. The largest differences were -0.54 ppbv for acetone at Deyang, -0.58 ppbv for acetaldehyde at Chengdu, 15 and -0.5 ppbv for acetone at Meishan, respectively. Based on the initial concentrations, the top three species contributing to $O_3$ formation potential were cis-2-butene, isoprene, m,p-xylene at Deyang; m,p-xylene, isoprene, acetaldehyde at Chengdu; and isoprene, ethylene, acetaldehyde at Meishan, respectively. These results differed from those based on observed concentrations. Comprehensively calculating the initial concentrations of VOCs helps accurately identify the key VOC species influencing $O_3$ formation.





20  **Abstract Art**

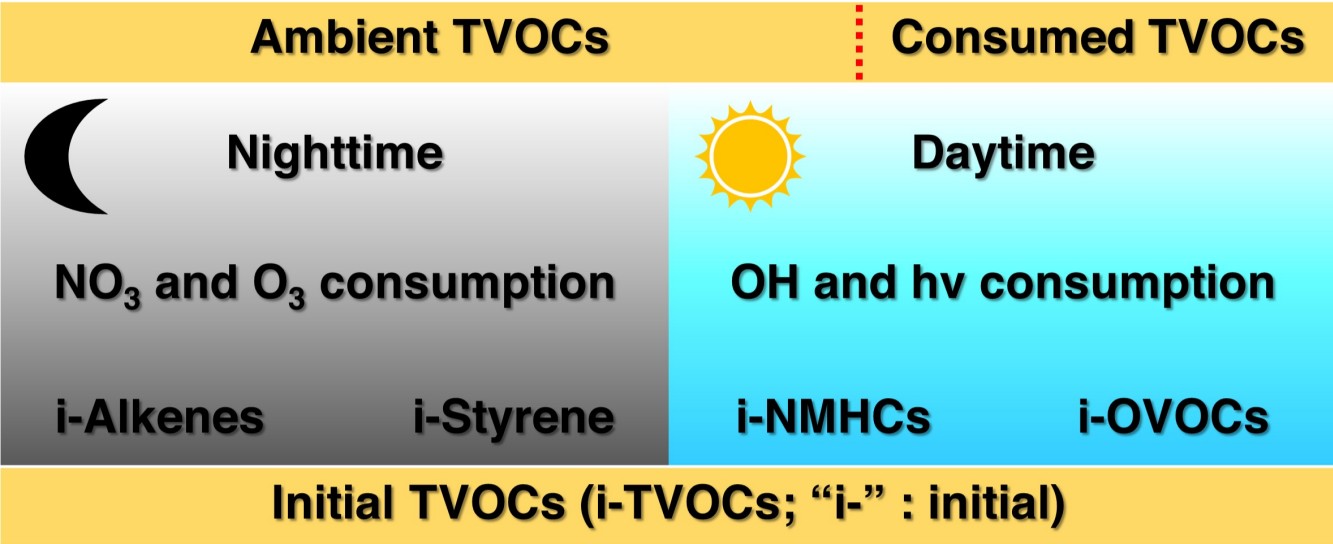

A schematic map showing differences between initial and ambient total VOCs (TVOCs) concentrations. Ambient TVOCs concentrations were measured in the sampling sites.



## 1 Introduction

Volatile organic compounds (VOCs) were key species in ozone ($O_3$) formation (Finlayson-Pitts and Pitts, 2000; Fry et al., 2018; Haagen-Smit and Fox, 1956; Seinfeld and Pandis, 2016). They mainly included non-methane hydrocarbons (NMHCs) and oxygenated VOCs (OVOCs) (Gkatzelis et al., 2021; Huang et al., 2020; Mo et al., 2021; Xia et al., 2020). VOCs primarily originated from anthropogenic sources such as biomass burning, vehicle emissions, solvent use, and industrial activities, as well as biogenic sources (Wu and Xie, 2017). The VOCs emitted from these sources were referred to as initial VOCs. Additionally, OVOCs could be formed through the oxidation of NMHCs (Birdsall and Elrod, 2011; Calvert et al., 2015; Finlayson-Pitts and Pitts, 2000). To more effectively develop VOCs emission reduction strategies, it was crucial to accurately identify the key initial VOC species that influenced $O_3$ formation.

The evaluation of the contribution of VOC species to $O_3$ typically involved methods such as the photochemical ozone creation potentials (Derwent et al., 1998; Derwent et al., 2007), the relative incremental reactivity method (Cardelino and Chameides, 1995), ozone formation pathway tracking based on box models (Zhan et al., 2023), and maximum incremental reactivity (MIR) method (Carter, 1994; Carter et al., 1995). Since 1994, the MIR method has been widely used (Carter, 1994; Carter et al., 1995; Kong et al., 2023). The MIR values were obtained by adding VOCs in the smog chambers and performing numerical simulations. These values reflected the changes in $O_3$ concentrations to add a unit of VOCs (Carter, 1994; Carter et al., 1995). The importance of VOCs species was ranked based on the product of MIR and the observed VOC concentrations at sampling sites to identify the key species contributing to $O_3$ formation potential (OFP) in Chengdu (Deng et al., 2019; Kong et al., 2023; Song et al., 2018; Tan et al., 2020a; Tan et al., 2020b; Wang et al., 2023; Xiong et al., 2021) and Deyang (Chen et al., 2021). For instance, the top three VOCs species of OFP at Chengdu were ethylene, propylene, and m,p-xylene (Song et al., 2018). However, the VOCs added in smog chambers should correspond to the initial VOCs emitted into the atmosphere from sources. Most studies relied on observed VOC concentrations. There were substantial uncertainties in accurately identifying key VOC species.

The ambient concentrations of VOCs were comprised of the remaining concentrations after the initial VOCs were consumed during the day and night, along with the secondary OVOCs concentrations during the day. Most studies calculated initial concentrations based on the daytime hydroxyl radicals (OH) consumption of NMHCs (He et al., 2019; Shao et al., 2009). However, the consumption of alkenes at night by $O_3$ or nitrate radicals ($NO_3$) and the primary emissions and secondary formation of OVOCs were not considered. First, although OH mainly reacted with NMHCs during the day, the nighttime consumption of alkenes was majorly driven by $NO_3$ or $O_3$ (Brown and Stutz, 2012; Finlayson-Pitts and Pitts, 1997; Zhu et al., 2020). The ratio of nighttime $NO_3$ and $O_3$ consumption to daytime OH consumption was $1.28\pm1.28$ for isoprene, styrene, 1,3-butadiene, ethylene, propylene, 1-butene, cis-2-butene, and trans-2-butene (de Gouw et al., 2017). Therefore, if nighttime alkene consumption was not considered, the initial concentrations of alkenes were likely underestimated.

Second, the observed concentrations of OVOCs included both primary emissions and secondary formation. The OFP of secondary OVOCs concentrations was already included in that of their precursor NMHCs because MIR represented the





maximum values. Thus, without excluding the secondary OVOCs formation, their OFP was likely overestimated. The differences in reaction rates of VOCs and secondary formation of OVOCs caused discrepancies in the species rankings of initial and observed VOC concentrations, which in turn affected the rankings of key VOCs species in OFP. Due to the instantaneous emissions, mixing, and oxidation changes in the ambient atmosphere, directly obtaining the initial VOC

concentrations was challenging during the day and night. Consequently, assessing initial VOC concentrations remained difficult. These caused uncertainties in accurately identifying key initial VOC species affecting $O_3$ formation.

To address these issues, this study calculated the initial concentrations of alkenes at night and NMHCs and OVOCs during the day based on reaction rates and observed VOC concentrations. It assessed the differences in VOCs species rankings of OFP based on initial and observed VOCs concentrations. These investigations were conducted by hourly measurements of 99

VOCs concentrations at rural Deyang, suburban Chengdu, and forest Meishan in the Sichuan Basin, China, from August to September 2019. The main aim was to constrain the consumption of VOCs at night and during the day, and the secondary formation of OVOCs during the day, to determine the initial VOCs concentrations and identify the key species affecting $O_3$ formation.

## 2 Materials and methods

### 2.1 Field measurements

Three monitoring sites were located in the Sichuan Basin, China (Fig. S1). One was located at Guihong Village (104º12' E, 31º1' N) in Deyang (expressed as rural Deyang), another was at Huangjueshu community (103º51' E, 30º24' N) in Chengdu (suburban Chengdu), and the third was at bamboo forest wetland park (103º49' E, 29º48' N) in Meishan (forest Meishan). The temperature ranged from 25 ºC to 35 ºC, and the relative humidity was from 40% to 80% in summer (July to September). First,

located between the eastern Tibetan Plateau and the western Longquan Mountain (Fig. S1b), air masses in the Deyang-Chengdu-Meishan urban agglomerations were not easily dispersed and tended to concentrate within the basin (Fig. S1c). Second, strong ultraviolet light, high temperatures, and low wind speeds were conducive to VOCs emissions and consumption during mid-latitude summer in the basin.

Hourly VOCs samples were collected, preprocessed, and analyzed by a custom-built online gas chromatography-mass

spectrometry/flame ionization detector (GC-MS/FID) system (TH-PKU 300B, Tianhong, China; GCMS-QP2010SE-Plus, Shimadzu, Japan) from 8 August to 14 September 2019. After moving particulate matters, moisture, and carbon dioxide, 300 mL of air was concentrated within electronic refrigeration at -150 ºC for each sample. The concentrated VOCs were quickly desorbed by heating at 100 ºC and transferred to the gas chromatographic column. $C_2$ to $C_5$ hydrocarbons were separated on a nonpolar capillary column (PLOT-$Al_2O_3$, Agilent, USA) and measured with the FID. Other compounds were separated on a

semi-polar column (DB-624, Agilent, USA) and detected by the MS. Standard curves were established for six concentrations from 0.1 ppbv to 8 ppbv for 57 photochemical assessment monitoring stations (PAMS) VOCs and 55 customized OVOCs and halocarbons with four internal standard gases (Linde, the United States). Data validation was performed almost every night





using the 2 ppbv PAMS and customized gas. The coefficients of determination of the calibration curves were greater than 0.99, and the method detection limits (MDLs) ranged from 0.003 ppbv to 0.070 ppbv for each VOC. A total of 99 VOCs were detected at the three sites, including 29 alkanes, 11 alkenes, 16 aromatics, 28 halocarbons, 13 OVOCs, acetonitrile, and acetylene.

## 2.2 The calculations of initial VOCs concentrations

Given the absence of secondary formation, VOCs other than oxygenated VOCs (OVOCs) were classified as NMHCs, which include alkanes, alkenes, acetylene, acetonitrile, halocarbons, and aromatics. Similarly, no secondary formation was found for methyl tert-butyl ether.

The major atmospheric oxidants for the consumption of VOCs were $NO_3$, $O_3$, and OH. Due to the absence of sunlight, OH concentrations and photolysis rates were very low at night. The $NO_3$ and $O_3$ were the primary oxidants for alkene and styrene consumption. The nighttime initial concentrations of alkenes and styrene were estimated using $NO_3$ or $O_3$ exposure methods, while other VOCs were excluded from the analysis due to their slow reaction rates with $NO_3$ and $O_3$ during nighttime.

During the daytime, $NO_3$ was highly unstable and rapidly photolyzed. Therefore, VOCs consumption by its oxidation was negligible. Alkenes and styrene can react with both OH and $O_3$. For alkenes and styrene, the ratio of the product of the OH reaction rates (Carter, 2010) and the observed OH concentration in the Chengdu Plain ($6.14 \times 10^6$ molecules cm$^{-3}$; Yang et al., 2021) to the product of the $O_3$ reaction rates (Atkinson and Arey, 2003; Carter, 2010) and measured $O_3$ concentrations (45.71 ppbv ) was 19.20. This indicated that VOCs were predominantly consumed by OH and $hv$ during daytime. The hourly $O_3$ concentrations were monitored concurrently by the Sichuan Environmental Monitoring Center at Deyang (104°26' E, 31°8' N), Chengdu (104°8' E, 30°37' N), and Meishan (103°52' E, 30°4' N) under a similar surrounding environment near the three VOC sampling sites, respectively. The daytime initial concentrations of VOCs, including methyl tert-butyl ether and NMHCs except for isoprene, were quantified using the OH exposure method. Further, initial concentrations of isoprene were calculated using the OH exposure method and its secondary products, methyl vinyl ketone (MVK) and methacrolein (MACR). Differently, initial concentrations of OVOCs were determined using the photochemical age method during the daytime, due to its primary emission, secondary production, and consumption by both OH and $hv$. Among meteorological factors, temperature was the primary driver of $O_3$ production (Jacob and Winner, 2009), with the OH reaction rate showing small variations between 25°C and 35°C. For instance, the reaction rate ratio for isoprene at these two temperatures was 0.96 (Saunders et al., 2003). Consequently, all VOC reaction rate constants were adjusted for a temperature of 300 K.

### 2.2.1 The calculations of initial NMHCs concentrations during nighttime

During local nighttime from 20:00 to 06:00 (Fig. S2), initial alkene and styrene concentrations were estimated through the $NO_3$ or $O_3$ exposure methods according to the relative loss rates of reported species between $NO_3$ and $O_3$ in the Los Angeles Basin (de Gouw et al., 2017). Unreported alkenes were classified through comparison with reported alkenes in reaction rates of both $NO_3$ ($kNO3$) and $O_3$ ($kO3$) (Fig. S3). For example, the nocturnal consumption of 1-butene was over 96% through



reaction with $O_3$ (de Gouw et al., 2017). The *kO3* for 1-pentene was higher than *kO3* for 1-butene, but *kNO3* for 1-pentene was lower than *kNO3* for 1-butene. Therefore, the initial 1-pentene concentrations were estimated using the $O_3$ exposure method. Briefly, isoprene, styrene, and 1,3-butadiene were grouped in the $NO_3$ exposure method, and 8 of 10 alkenes in the $O_3$ exposure method. For $NO_3$ consumption, the initial concentrations of each alkene ([i-alkene]) were estimated (de Gouw et al., 2017):

$$[NO_3]\Delta t = \frac{1}{(kNO3_{benzene} - kNO3_{isoprene})} \times \left[ \ln\left(\frac{[\text{i-benzene}]}{[\text{i-isoprene}]}\right) - \ln\left(\frac{[\text{m-benzene}]}{[\text{m-isoprene}]}\right) \right] \tag{Eq. 1}$$

$$[\text{i-alkene}_j] = [\text{a-alkene}_j] \times \exp(kNO3_{alkene_j}[NO_3]\Delta t) \tag{Eq. 2}$$

where $[NO_3]$ and $\Delta t$, together referred to as $NO_3$ exposure ($[NO_3]\Delta t$), were the concentrations of $NO_3$ and nocturnal reaction time, respectively. $kNO3_{benzene}$ and $kNO3_{isoprene}$ were the reaction rate constants of benzene $(3.0 \times 10^{-17}$ $cm^3$ $molecule^{-1}$ $s^{-1})$ and isoprene $(6.8 \times 10^{-13}$ $cm^3$ $molecule^{-1}$ $s^{-1})$ with $NO_3$, respectively (Atkinson and Arey, 2003; Carter, 2010). [i-benzene]/[i-

isoprene] values represented the initial emission ratios between benzene and isoprene. The initial emission ratios were estimated at $0.5 \pm 0.3$ppbv ppbv$^{-1}$ at Deyang, $0.5 \pm 0.3$ppbv ppbv$^{-1}$ at Chengdu, and $0.2 \pm 0.1$ppbv ppbv$^{-1}$ at Meishan from measured data with a low degree of nocturnal consumption, respectively (Fig. S4). The emission ratios were directly linked to emission sources. After mixing from different sources, the emission ratios obtained at different sampling sites may vary. Although benzene and isoprene originated from different sources, anthropogenic activities in the Chengdu Plain were relatively

stable at night based on positive matrix factorization results (Kong et al., 2023; Xiong et al., 2021). Therefore, their initial emission ratios may remain consistent for each source. [m-benzene]/[m-isoprene] were the ratios of measured hourly concentrations between benzene to isoprene. [m-alkenes$_j$] and $kNO3_j$ referred to the observed concentrations and $NO_3$ reaction rate constants (Fig. S3) of species $j$ in isoprene, styrene, and 1,3-butadiene, respectively.

For $O_3$ consumption, the initial concentrations of 8 reactive alkenes were estimated (de Gouw et al., 2017):

$$[O_3]\Delta t = \frac{1}{(kO3_{benzene} - kO3_{cis-2-butene})} \times \left[ \ln\left(\frac{[\text{i-benzene}]}{[\text{i-cis-2-butene}]}\right) - \ln\left(\frac{[\text{m-benzene}]}{[\text{m-cis-2-butene}]}\right) \right] \tag{Eq. 3}$$

$$[\text{i-alkene}_j] = [\text{a-alkene}_j] \times \exp(kO3_{alkene_j}[O_3]\Delta t) \tag{Eq. 4}$$

where $[O_3]$ and $\Delta t$ were referred to as $O_3$ exposure ($[O_3]\Delta t$) together. $kO3_{benzene}$ and $kO3_{cis-2-butene}$ were the reaction rate constants of benzene $(1.0 \times 10^{-20}$ $cm^3$ $molecule^{-1}$ $s^{-1})$ and cis-2-butene $(1.3 \times 10^{-16}$ $cm^3$ $molecule^{-1}$ $s^{-1})$ with $O_3$ (Atkinson and Arey, 2003; Carter, 2010). [i-benzene]/[i-cis-2-butene] values represented the initial emission ratios between benzene and cis-2-butene.

The estimated initial emission ratios were $0.5 \pm 0.3$ppbv ppbv$^{-1}$ at Deyang, $4.5 \pm 1.0$ppbv ppbv$^{-1}$ at Chengdu, and $6.5 \pm 1.0$ppbv ppbv$^{-1}$ at Meishan from measured data with a low degree of nocturnal consumption, respectively (Fig. S5). Similar to initial emission ratios of benzene to isoprene, initial emission ratios of benzene to cis-2-butene may remain consistent for each source. After mixing from different sources, the emission ratios obtained at different sampling sites may vary. [m-benzene]/[m-cis-2-butene] were the ratios of measured hourly concentrations between benzene and cis-2-butene. [m-alkenes$_j$] and $kO3_j$ referred

to the observed hourly concentrations and $O_3$ reaction rate constants (Fig. S3) of the species $j$ in alkenes, respectively. The cis-2-butene was replaced with trans-2-butene at Chengdu, due to the unavailability of cis-2-butene data.



### 2.2.2 The calculations of initial NMHCs concentrations during daytime

During local daytime from 7:00 to 19:00 (Fig. S2), the initial concentrations of each NMHC [i-NMHC] were estimated, besides methyl tert-butyl ether (de Gouw et al., 2005; Ma et al., 2022; Roberts et al., 1984; Shao et al., 2011):

$$[OH]\Delta t = \frac{1}{(kOH_{ethylbenzene} - kOH_{m,p\text{-}xylenes})} \times \left[\ln\left(\frac{[\text{i-ethylbenzene}]}{[\text{i-m,p-xylenes}]}\right) - \ln\left(\frac{[\text{m-ethylbenzene}]}{[\text{m-m,p-xylenes}]}\right)\right] \tag{Eq. 5}$$

$$[\text{i-NHMC}_j] = [\text{m-NHMC}_j] \times \exp(kOH_{NMHC_j}[OH]\Delta t) \tag{Eq. 6}$$

where [OH] and $\Delta t$ were referred to together as OH exposure ($[OH]\Delta t$). [i-ethylbenzene]/[i-m,p-xylenes] values represented the initial emission ratios between ethylbenzene and m,p-xylenes (Fig. S7). The major source of ethylbenzene and m,p-xylenes in the Chengdu Plain was solvent use (Wu and Xie, 2017). A good linear correlation in concentrations between ethylbenzene and m,p-xylenes was observed ($R^2 = 0.96$). $kOH_{ethylbenzene}$ and $kOH_{m,p\text{-}xylenes}$ were the reaction rate constants of ethylbenzene ($7.0 \times 10^{-12}$ cm$^3$ molecule$^{-1}$ s$^{-1}$) and m,p-xylenes ($1.9 \times 10^{-11}$ cm$^3$ molecule$^{-1}$ s$^{-1}$) with OH, respectively (Carter, 2010). Therefore, we selected them to calculate OH exposure. [m-NMHC$_j$] and $kOH_j$ denoted the observed hourly concentrations and OH reaction rate constants (Fig. S6) of the species $j$ in NMHCs, respectively.

The $[OH]\Delta t$ for initial concentrations of isoprene ([i-isoprene]) was estimated directly through its photochemical products, MVK and MACR (Paulot et al., 2009; Stroud et al., 2001).

### 2.2.3 The calculations of initial OVOCs concentrations during daytime

During the daytime, initial concentrations of MVK and MACR ([i-OVOC]) were estimated:

$$[\text{c-isoprene}] = [\text{i-isoprene}] - [\text{m-isoprene}] \tag{Eq. 7}$$

$$[\text{s-OVOC}_j] = p \times [\text{c-isoprene}] \tag{Eq. 8}$$

$$[\text{i-OVOC}_j] = [\text{m-OVOC}_j] - [\text{s-OVOC}_j] + [\text{c-isoprene}] \times \frac{([\text{m-OVOC}_j] - [\text{s-OVOC}_j]) \times kOH^*_{OVOC_j}}{[\text{m-isoprene}] \times kOH_{isoprene}} \tag{Eq. 9}$$

where [c-isoprene] indicated consumed concentrations of isoprene, which were equal to its initial concentrations ([i-isoprene]) calculated from Eq. (6) minus the measured concentrations ([m-isoprene]). [s-OVOC$_j$] represented the secondary concentrations of species $j$ in MVK or MACR produced from isoprene oxidation. The p values in Eq. (8) represented the molecular production from one molecular unit of isoprene consumption, with 0.32 for MVK and 0.23 for MACR, respectively (Paulot et al., 2009). Because the values of $kOH_{MVK}$ and $kOH_{MACR}$ values were 3.5 times and 5 times lower than $kOH_{isoprene}$ (Fig. S6), respectively, the measured concentrations ([m-OVOC$_j$]) were assumed to be instantaneous total concentrations in Eq. (9). Therefore, [i-OVOC$_j$] was equal to [m-OVOC$_j$] minus [s-OVOC$_j$] and then plus the corresponding photochemical consumption approximately, which was calculated using [c-isoprene] and reaction rates. The photochemical consumption of OVOCs included both photolysis and reaction with OH. Therefore, the total OVOC$_j$ loss rates ($kOH^*_{ovocj}$) were estimated based on the photolysis rate ($J_{OVOCj}$) and loss rate with OH ($[OH]kOH_{OVOCj}$). The ratios of $J_{NO2}$ to OH concentrations in the Sichuan Basin were similar to those in the Los Angeles Basin (de Gouw et al., 2018; Yang et al., 2021), so the $J_{OVOCj}$ and $[OH]kOH_{OVOCj}$ may be comparable (de Gouw et al., 2018; Tan et al., 2018). Accordingly, we assumed that the ratios (0.6 for MVK and MACR)



of $J_{OVOCj}$ to $[OH]kOH_{OVOCj}$ established in the Los Angeles Basin (Fig. S8; de Gouw et al., 2018) were applicable for estimating $kOH^*_{ovocj} = (1 + 0.6) \times kOH_{OVOCj}$ in the Sichuan Basin. $kOH_{isoprene}$ meant the OH reaction rate constant of isoprene.

To differentiate the secondary production and consumption, we estimated the initial concentrations [i-OVOC] of OVOCs during daytime through the photochemical age method, except for methyl tert-butyl ether, MVK, and MACR (de Gouw et al., 2005; de Gouw et al., 2018; Wu et al., 2020):

$$[\text{m-OVOC}_j] = ER_{OVOC_j} \times [\text{m-benzene}] \times \exp(-(kOH^*_{OVOC_j} - kOH_{benzene})[OH]\Delta t) +$$

$$ER_{HC} \times [\text{m-benzene}] \times \frac{kOH_{HC}}{kOH^*_{OVOC_j} - kOH_{HC}} \times \frac{\exp(-kOH_{HC}[OH]\Delta t) - \exp(-kOH^*_{OVOC_j}[OH]\Delta t)}{\exp(-kOH_{benzene}[OH]\Delta t))} +$$

$$ER_{biogenic} \times [\text{i-isoprene}] \qquad\qquad\qquad\qquad\qquad\qquad\qquad\qquad\qquad\qquad\qquad\qquad\text{(Eq. 10)}$$

$$[\text{i-OVOC}_j] = ER_{OVOC_j} \times [\text{m-benzene}] + ER_{biogenic} \times [\text{i-isoprene}] \qquad\qquad\qquad\qquad\text{(Eq. 11)}$$

where the measured concentrations of species $j$ in OVOCs ([m-OVOC$_j$]) equaled the sum of primary anthropogenic contributions, secondary anthropogenic contributions, and biogenic contributions, as represented sequentially in Eq. (10). Benzene was selected as the tracer of anthropogenic primary sources due to dominant combustion and industrial VOCs

emissions in the Sichuan Basin (Wu and Xie, 2017) and its relatively low OH reaction rate. $ER_{OVOCj}$ and $ER_{HC}$ were the emission ratios of species $j$ in OVOCs and hydrocarbons to benzene, respectively. We assumed that the ratios (R) for $J_{OVOCj}$ to $[OH]kOH_{OVOCj}$ established in the Los Angeles Basin (Fig. S8; de Gouw et al., 2018) were applicable for estimating $kOH^*_{ovocj} = (1 + R) \times kOH_{OVOCj}$ in the Sichuan Basin. OH exposure was calculated by Eq. (5). $ER_{biogenic}$ values represented the initial emission ratios between OVOCs and isoprene from biogenic sources. [i-isoprene] were estimated by Eq. (6). The $ER_{OVOCj}$,

$ER_{HC}$, $kOH_{HC}$, and $ER_{biogenic}$ were determined using the nonlinear least-squares fit.

**2.3 O₃ formation potential**

OFP was calculated using in Eq. (12) or Eq. (13):

$$[\text{OFP}] = MIR_j \times [\text{VOCs}_j] \qquad\qquad\qquad\qquad\qquad\qquad\qquad\qquad\qquad\qquad\qquad\text{(Eq. 12)}$$

$$[\text{OFP}] = MIR_j \times [\text{i-VOCs}_j] \qquad\qquad\qquad\qquad\qquad\qquad\qquad\qquad\qquad\qquad\text{(Eq. 13)}$$

where $MIR_j$, [VOCs$_j$], and [i-VOCs$_j$] indicated the maximum incremental reactivity (Carter, 2010), measured, and initial concentrations of species $j$ in VOCs, respectively.



## 3 Results and discussion

Figure 1 shows the hourly ambient total VOCs (TVOCs) concentrations at rural Deyang, suburban Chengdu, and forest Meishan from August to September 2019. The average TVOCs concentration was 37.63±15.34 ppbv (parts per billion by volume) at Deyang, which was higher than 33.42±13.96 ppbv at Chengdu and 25.17±8.46 ppbv at Meishan (Figure 1).

**Fig. 1.** Hourly concentrations of ambient VOCs at Deyang (a), Chengdu (b), and Meishan (c) from August to September 2019, respectively. The lack of data at rural Deyang from 30 August to 4 September 2019 was due to a power failure.

Due to the influences of sunlight, the consumption of initial VOCs was classified into nighttime and daytime periods.
During the night, initial alkenes and styrene were consumed through oxidation by $NO_3$ radicals or $O_3$ (Calvert et al., 2015; de Gouw et al., 2017; Finlayson-Pitts and Pitts, 2000). Because the reaction rates of $NO_3$ and $O_3$ were very low, mostly below $10 \times 10^{-21}$ cm$^3$ molecule$^{-1}$ s$^{-1}$ (Atkinson, 2000; Atkinson and Arey, 2003; Carter, 2010), the nighttime consumption of aromatics



except for styrene, alkanes, OVOCs, halocarbons, acetonitrile, and acetylene was not considered. During the day, initial NMHCs were primarily consumed by OH. Initial OVOCs were consumed by OH and photolysis during the daytime, after excluding secondary formation.

The largest difference was found at Deyang among the three sites between the initial and observed average concentrations of TVOCs, with a difference of 3.10 ppbv (Table 1). Due to the high reactivities (Carter, 2010), alkenes and aromatics were mainly responsible for this difference. The average initial concentrations of alkenes were 6.40 ppbv at Deyang, 5.24 ppbv at Chengdu, and 6.21 ppbv at Meishan, respectively, which were higher than the observed concentrations of 3.32 ppbv, 2.96 ppbv, and 3.89 ppbv, respectively (Table 1). The average initial concentrations of aromatics were 3.92 ppbv, 4.97 ppbv, and 2.15 ppbv, which were higher than the observed concentrations of 2.99 ppbv, 4.04 ppbv, and 1.92 ppbv, respectively (Table 1). Among different chemical groups at the three sites, the largest difference was 3.08 ppbv for alkenes at Deyang (Table 1). This difference was attributed to higher initial alkene concentrations around Deyang (Table 1) and, consequently, higher consumption of alkenes. Specifically, the largest differences were 1.04 ppbv for cis-2-butene at Deyang, 0.86 ppbv for isoprene at Chengdu, and 1.98 ppbv for isoprene at Meishan, respectively (Figure 2, Tables S1, and S2). The reaction rates of isoprene with $NO_3$ and OH were higher than those of cis-2-butene, but the largest difference was for cis-2-butene at Deyang. This was possibly due to higher initial concentrations of cis-2-butene compared to those of isoprene (Figure 2, Tables S1, and S2), which led to greater consumption and thus larger differences for cis-2-butene at Deyang.

Table 1 The ambient and initial concentrations of VOCs groups at the same time from August to September 2019.

| Groups (Mean ± SD) | Deyang | | Chengdu | | Meishan | |
|---|---|---|---|---|---|---|
| | [VOCs] | [i-VOCs] | [VOCs] | [i-VOCs] | [VOCs] | [i-VOCs] |
| TVOCs | 37.31 ± 15.16 | 40.41 ± 16.52 | 33.12 ± 13.93 | 35.36 ± 14.50 | 25.13 ± 8.54 | 26.47 ± 9.07 |
| Alkanes | 12.36 ± 6.47 | 12.71 ± 6.45 | 10.19 ± 5.33 | 10.48 ± 5.32 | 7.61 ± 2.79 | 7.70 ± 2.83 |
| Alkenes | 3.32 ± 2.25 | 6.40 ± 4.32 | 2.96 ± 1.65 | 5.24 ± 3.35 | 3.89 ± 3.08 | 6.21 ± 3.88 |
| Acetylene and Acetonitrile | 2.87 ± 2.05 | 2.90 ± 2.07 | 2.52 ± 1.39 | 2.54 ± 1.39 | 1.99 ± 0.76 | 2.00 ± 0.76 |
| Halocarbons | 4.89 ± 2.74 | 4.90 ± 2.75 | 4.93 ± 3.56 | 4.94 ± 3.56 | 3.05 ± 1.42 | 3.06 ± 1.43 |
| Aromatics | 2.99 ± 2.49 | 3.92 ± 3.18 | 4.04 ± 3.20 | 4.97 ± 3.59 | 1.92 ± 1.01 | 2.15 ± 1.12 |
| OVOCs | 10.89 ± 5.96 | 9.57 ± 4.02 | 8.48 ± 5.93 | 7.19 ± 3.84 | 6.67 ± 4.65 | 5.36 ± 3.38 |

The differences between the initial and observed concentrations of aromatics were 0.93 ppbv at both Deyang and Chengdu and 0.23 ppbv at Meishan (Table 1). Overall, the initial concentrations of aromatics at Chengdu were higher than those at Deyang, which likely led to higher aromatics consumption. However, the larger consumption of styrene, a reactive aromatic, at Deyang resulted in equal consumption of aromatics at 0.93 ppbv at both Deyang and Chengdu (Table 1). Styrene exhibited



the largest difference among aromatics between initial and observed concentrations, with differences of 0.59 ppbv at Deyang, 0.30 ppbv at Chengdu, and 0.10 ppbv at Meishan, respectively (Figure 2, Tables S1 and S2). This was due to its high OH reaction rate, and its consumption during nighttime reactions with $NO_3$. For aromatics, the second-largest differences at Chengdu were 0.28 ppbv for m,p-xylene (Figure 2, Tables S1 and S2). This was possibly due to higher emissions of solvent use around Chengdu compared to the other two sites, which likely caused higher consumption and thus higher differences.

Due to secondary formation, the average initial OVOCs concentrations were lower than the observed concentrations by 1.32 ppbv at Deyang, 1.29 ppbv at Chengdu, and 1.31 ppbv at Meishan, respectively (Table 1). Specifically, acetone showed the largest difference between initial and observed concentrations, with differences of -0.54 ppbv at Deyang and -0.50 ppbv at Meishan, respectively (Figure 2, Tables S1 and S2). Acetaldehyde showed the largest difference at -0.58 ppbv at Chengdu (Figure 2, Tables S1 and S2). This might be due to the higher emissions of isoalkanes at Deyang. They could convert into acetone (Jacob et al., 2002), which thus caused a larger negative difference. Due to the low reaction rates of OH, $NO_3$, and $O_3$ (Carter, 2010), the initial and observed concentrations of alkanes, acetylene, acetonitrile, and halocarbons were almost the same (Table 1). Overall, major differences between initial and observed concentrations were found in alkenes, aromatics, and OVOCs, due to high reactivities and secondary OVOCs formation.





**Fig. 2.** The ambient (a) and initial (b) VOCs concentrations at Deyang, Chengdu, and Meishan from August to September 2019,

255   respectively. Square dots and whiskers showed the mean and standard error (SE) values, respectively. Details were shown in Table S1.



Concentrations of alkanes were the largest among different VOCs chemical groups (Figure 1). However, due to their relatively low reaction activities, lower OFP values and smaller differences between observed and initial concentrations were found (Figure 3). There were significant differences in OFP between initial and observed alkene concentrations. The top three chemical groups with the largest differences included alkenes, aromatics, and OVOCs at all three sites. However, the OFP rankings of chemical groups differed between initial and observed concentrations. The top three chemical groups in OFP based on observed concentrations were OVOCs, aromatics, and alkenes at Deyang. Based on initial concentrations, the top three were alkenes, aromatics, and OVOCs. The top three based on observed concentrations were aromatics, oxygenated VOCs, and alkenes at Chengdu, while based on initial concentrations, they were aromatics, alkenes, and OVOCs. The top three based on observed concentrations were alkenes, OVOCs, and aromatics at Meishan, while based on initial concentrations, they were alkenes, aromatics, and OVOCs. The OFP of alkenes based on initial concentrations was highest at Meishan among the three sites, reaching 141.90 µg m$^{-3}$. The OFP of alkenes based on initial concentrations was higher than that of aromatics at both Deyang and Meishan, but lower at Chengdu. This might be due to the higher emissions of solvents around Chengdu, leading to higher initial concentrations of aromatics (Table 1) and thus the higher OFP compared to alkenes.

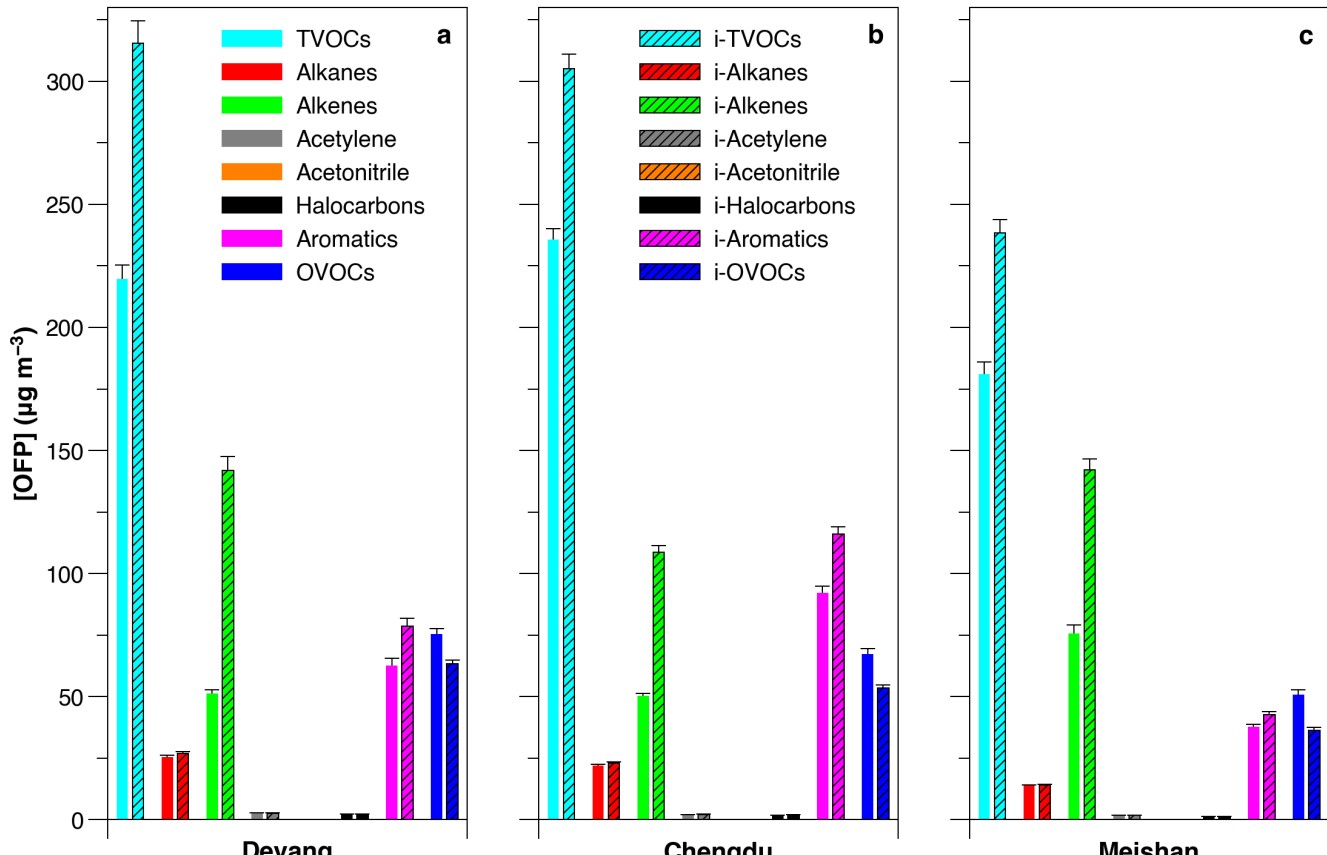

**Fig. 3.** OFP values based on ambient and initial VOCs concentrations at Deyang, Chengdu, and Meishan, from August to September 2019, respectively. Mean ± SE was shown.



The temporal variation trends of the OFP were relatively consistent at the three sites based on initial and observed TVOC concentrations (Figure S9ab). The OFP based on initial and observed TVOC concentrations ranged from 76.81 μg m$^{-3}$ to 1476.28 μg m$^{-3}$ and from 55.86 μg m$^{-3}$ to 827.30 μg m$^{-3}$, respectively at Deyang. There were the most significant variations in alkenes among chemical groups, with ranges of 22.22 μg m$^{-3}$ to 1081.41 μg m$^{-3}$ and 7.87 μg m$^{-3}$ to 282.78 μg m$^{-3}$, respectively.
The highest OFP based on the initial TVOC concentrations was found at 16:00 on August 11, 2019, primarily due to the contribution of 1081.41 μg m$^{-3}$ from alkenes. At this time, the OFP based on the observed TVOC concentrations was only 327.41 μg m$^{-3}$. The ratio of the OFP between the initial and observed TVOC concentrations exceeded 4.5. This discrepancy was because alkenes had been significantly consumed at that time. If the consumption of alkenes had not been considered, their contributions to the OFP would have been greatly underestimated.

The OFP based on initial and observed TVOCs concentrations ranged from 48.61 μg m$^{-3}$ to 1143.74 μg m$^{-3}$ and from 39.21 μg m$^{-3}$ to 819.61 μg m$^{-3}$, respectively at Chengdu (Figure S9cd). The greatest variations were found in alkenes among chemical groups, with ranges from 14.26 μg m$^{-3}$ to 780.85 μg m$^{-3}$ and from 4.94 μg m$^{-3}$ to 336.58 μg m$^{-3}$, respectively. The highest OFP based on the initial TVOC concentrations was found at 12 pm on August 30, 2019, primarily due to contributions of 500.64 μg m$^{-3}$ from alkenes and 494.89 μg m$^{-3}$ from aromatics. At this time, the OFP based on the observed TVOC concentrations was 819.61 μg m$^{-3}$. The OFP based on the initial and observed TVOC concentrations were quite close, indicating that photochemical consumption was relatively low compared to that at Deyang. Furthermore, the OFP of observed OVOCs concentrations was 171.94 μg m$^{-3}$, higher than the OFP of initial concentrations at 88.40 μg m$^{-3}$. Compared to the other two sites, emissions from solvent use were higher around Chengdu. This led to higher emissions of aromatics, which significantly contributed to the OFP.

The ranges of OFP based on initial and observed TVOCs concentrations were from 65.34 μg m$^{-3}$ to 1351.58 μg m$^{-3}$ and from 48.43 μg m$^{-3}$ to 1077.27 μg m$^{-3}$ (Figure S9). The OFP of alkenes showed the greatest variations, ranging from 22.60 μg m$^{-3}$ to 1133.53 μg m$^{-3}$ and from 7.62 μg m$^{-3}$ to 864.52 μg m$^{-3}$. The highest OFP of the initial TVOC concentrations was found at 18:00 on August 17, 2019, primarily due to the contribution of 1133.53 μg m$^{-3}$ from alkenes. Because of low photochemical consumption and secondary OVOCs formation, the difference in OFP between initial and observed TVOCs concentrations was relatively low, at about 25%, at this time. Compared to the other two sites, there were more biogenic isoprene emissions around Meishan, which contributed to the OFP of alkenes.

The highest hourly OFP values of TVOCs initial concentrations were 391.07 μg m$^{-3}$ at 12 pm at Deyang, 432.12 μg m$^{-3}$ at 12 pm at Chengdu, and 403.80 μg m$^{-3}$ at 6 pm at Meishan, respectively (Figure S10). The diurnal variations of OFP at Deyang and Chengdu were consistent with those of sunlight intensity. The OFP at Meishan was primarily contributed by isoprene. Bamboo around Meishan could emit isoprene. The accumulation of isoprene caused its highest concentrations at 6 pm and thus OFP.

According to Table 2, the number of species that ranked in the top ten for both observed and initial OFP concentrations were 8 for Deyang, 8 for Chengdu, and 9 for Meishan. Although the number of common species was high, their rankings differed. The importance of isoprene increased notably in OFP from observed to initial VOCs concentrations, rising from tenth





305 to second at Deyang, from sixth to second at Chengdu, and remaining first with more than a twofold increase at Meishan. The species with the highest OFP based on initial VOC concentrations was cis-2-butene, with a value of 39.49 μg m$^{-3}$ at Deyang, whereas this species ranked tenth based on observed concentrations. This indicated that not considering initial concentrations could underestimate its importance for O$_3$ formation. Four of the top ten OFP species based on initial concentrations were aromatics at Chengdu, suggesting the need to control solvent usage. Ioprene ranked first in OFP based on both observed and

310 initial OFP concentrations at Meishan. Its proportion increased due to relatively small changes in OFP for other species. The top three species of OFP based on initial VOCs concentrations were cis-2-butene, isoprene, m,p-xylene at Deyang; m,p-xylene, isoprene, acetaldehyde at Chengdu; and isoprene, ethylene, acetaldehyde at Meishan, respectively. These results differed from those based on observed concentrations (Table 2) and those reported at Chengdu from 2016 to 2017, which were ethylene, propylene, and m,p-xylene (Song et al., 2018).

315 The initial concentrations of VOCs were directly related to the MIR values. Many studies have evaluated key VOC species based on observed VOCs concentrations or only considering OH consumption of NMHCs. This study systematically calculated the initial concentrations of NMHCs and OVOCs during both day and night, which helps in accurately identifying the key VOC species affecting O$_3$ formation.



Table 2 Differences of top ten OFP species based on observed and initial VOCs concentrations.

| Sites | Top 10 species of OFP values (µg m$^{-3}$) | | | |
|---|---|---|---|---|
| | Based on observed [VOCs] | | Based on initial [VOCs] | |
| Deyang | Acetaldehyde | 29.37±18.3 | cis-2-Butene | 39.49±65.95 |
| | Ethylene | 22.22±16.95 | Isoprene | 35.24±21.43 |
| | m,p-Xylene | 21.77±27.93 | m,p-Xylene | 25.96±28.29 |
| | Hexanal | 14.0±15.5 | Acetaldehyde | 24.45±12.97 |
| | Toluene | 11.01±8.96 | Ethylene | 23.97±16.98 |
| | Propylene | 9.4±10.02 | trans-2-Butene | 18.08±31.9 |
| | o-Xylene | 9.14±11.92 | Hexanal | 14.72±11.79 |
| | Methyl vinyl ketone | 8.14±5.83 | Propylene | 12.37±10.65 |
| | Isoprene | 7.77±7.6 | Toluene | 11.72±9.05 |
| | cis-2-butene | 5.41±3.22 | o-Xylene | 10.43±12.12 |
| Chengdu | Acetaldehyde | 45.85±51.6 | m,p-Xylene | 53.37±48.73 |
| | m,p-Xylene | 44.04±44.02 | Isoprene | 39.33±29.31 |
| | o-Xylene | 17.29±17.33 | Acetaldehyde | 39.02±29.36 |
| | Ethylene | 16.53±10.8 | 1-Butene | 29.91±46.08 |
| | Toluene | 14.01±11.15 | o-Xylene | 19.99±18.88 |
| | Isoprene | 13.92±18.46 | Ethylene | 18.2±10.73 |
| | 1-Butene | 12.54±15.41 | Toluene | 15.06±11.77 |
| | Ethylbenzene | 6.03±5.65 | Propylene | 8.06±5.53 |
| | Propylene | 5.53±4.67 | Ethylbenzene | 6.55±5.95 |
| | Methyl vinyl ketone | 5.08±5.51 | 1,3,5-Trimethylbenzene | 5.75±14.77 |
| Meishan | Isoprene | 43.65±74.97 | Isoprene | 102.04±98.33 |
| | Acetaldehyde | 21.57±24.5 | Ethylene | 19.69±12.27 |
| | Ethylene | 18.9±12.27 | Acetaldehyde | 18.27±16.99 |
| | m,p-Xylene | 16.09±12.12 | m,p-Xylene | 18.03±12.64 |
| | Methyl vinyl ketone | 10.27±13.97 | Propylene | 7.29±7.01 |
| | o-Xylene | 6.68±4.94 | o-Xylene | 7.27±5.11 |
| | Propylene | 6.37±6.79 | Toluene | 5.65±3.92 |
| | Toluene | 5.46±3.84 | Methyl vinyl ketone | 4.61±7.09 |
| | Methacrolein | 4.44±4.92 | 1-Butene | 4.15±4.73 |
| | Hexanal | 3.14±1.56 | Hexanal | 2.9±1.42 |



 **4 Conclusions**

This study focused on comparing the differences between initial and ambient VOC concentrations, which were crucial for calculating OFP. Using reaction rates and hourly concentrations of 99 VOCs in the Chengdu Plain, China, this study calculated the initial concentrations of VOCs during both day and night in summer and compared them with observed concentrations. The results showed that the average initial concentrations of alkenes and aromatics were significantly higher than the observed concentrations. The largest differences between initial and measured concentrations were 1.04 ppbv for cis-2-butene at Deyang, 0.86 ppbv for isoprene at Chengdu, and 1.98 ppbv for isoprene at Meishan, respectively. Because of the secondary production, the initial OVOCs concentrations were lower than the observed ones. The largest differences were -0.54 ppbv for acetone at Deyang, -0.58 ppbv at Chengdu for acetaldehyde, and  -0.5 ppbv at Meishan for acetone, respectively. Based on the initial VOCs concentrations, the top three species contributing to OFP were cis-2-butene, isoprene, m,p-xylene at Deyang; m,p-xylene, isoprene, acetaldehyde at Chengdu; and isoprene, ethylene, acetaldehyde at Meishan, respectively. These results differed from those based on observed concentrations. Comprehensively calculating the initial concentrations of VOCs helps accurately identify the key VOC species contributing OFP.

**Associated content**

**Supporting information:**

The details of three VOCs monitoring sites, the calculation parameters for initial VOCs concentrations, and the details of OFP for VOCs chemical groups.

**Acknowledgements**

This study was supported by the National Key Research and Development Program of China (Grant 2018YFC02140001).

**Authors' contributions**

X.D.Z. conceived the idea for the protocol and experiment design under the guidance of S.D.X. All authors contributed to the collection of observations. X.D.Z. wrote the manuscript.

**Competing interests**

The authors declare no competing interests.



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
