# Peer review of "Differences in key volatile organic compound species in ozone formation between their initial and measured concentrations"

_EGUsphere, 2024_

## Author Comment (AC1)

**Authors' responses to review comments**

Atmospheric Chemistry and Physics (egusphere-2024-2568)

**Differences in the key volatile organic compound species between their emitted and ambient concentrations in ozone formation**

**Reviewer #2**

1. Zheng and Xie present a method for assessing the ozone forming potential of a series of volatile organic compounds measured in the Sichuan Basin, China. The method is based on observation of ambient concentrations of VOCs, followed by estimates of what is then referred to as "initial" VOCs, intended to represent the emitted amounts of these VOCs or the amount produced by secondary production in the atmosphere. Ozone formation potentials are then calculated using literature maximum incremental reactivities based on either observed or corrected VOCs.

   The manuscript is clearly written and the figures are of good quality for presentation. It will be of interest to the readership of ACP.

**Response:** Thank you for reviewing our manuscript.

2. The major comment is that the classification of VOCs by chemical functional group rather than sources may lead to some errors. This is especially true for alkenes, but may also pertain to the oxygenates. Anthropogenic alkenes should be treated differently from biogenic alkenes (mainly isoprene), since these VOCs have very different sources. Division into these categories would make the analysis methods also self-consistent, although it may require some change in methodology for isoprene itself.

**Response:**

We have revised the overall classification method in Section 2.2. The emitted VOC concentrations are classified into three categories: anthropogenic sources in Section 2.2.1, biogenic sources in Section 2.2.2, and a combination of both in Section 2.2.3. In addition to secondary production, ten of the 13 OVOCs (except MTBE, MVK, and MACR) are also emitted from both anthropogenic and biogenic sources (Zou et al., 2024; Lyu et al., 2024; Li et al., 2023; Wu et al., 2020). We revise the sentences (please check lines 97-245 in Materials and methods). We also update the comparison between emitted and ambient VOC concentrations, as well as their corresponding OFP values in Results and discussion.

*'2.2 Calculations of emitted VOCs concentrations*

*Source classification is crucial for calculating emitted VOCs concentrations. NMHCs (except isoprene) and MTBE are generally emitted from anthropogenic activities. Isoprene is typically emitted from biogenic sources and oxidized into methyl vinyl ketone (MVK) and methacrolein (MACR). In addition to secondary production, ten of the 13*

*OVOCs (except MTBE, MVK, and MACR) are also emitted from both anthropogenic and biogenic sources (Zou et al., 2024; Lyu et al., 2024; Li et al., 2023; Wu et al., 2020). The emitted VOC concentrations are classified into three categories: anthropogenic sources in Section 2.2.1, biogenic sources in Section 2.2.2, and a combination of both in Section 2.2.3.*

*First, the major atmospheric oxidants for the consumption of emitted VOCs are $NO_3$, $O_3$, and OH. Due to the absence of sunlight, OH concentrations and photolysis rates are very low from 20:00 to 06:00 (Fig. S2). Either $NO_3$ or $O_3$ are the primary oxidants for the consumption of emitted alkene and styrene during the nighttime. During the local nighttime, emitted alkene and styrene concentrations are estimated through the $NO_3$ or $O_3$ exposure methods based on the relative loss rates of reported species between $NO_3$ and $O_3$ in the Los Angeles Basin (de Gouw et al., 2017). Other VOCs are excluded from the analysis due to their slow reaction rates with $NO_3$ and $O_3$ during nighttime. The $NO_3$ or $O_3$ exposure method indicates that the concentration ratios of a stable tracer species to a reactive tracer species would increase with both $NO_3$ or $O_3$ concentrations and reaction time after emissions. Emitted concentrations are calculated based on $NO_3$ or $O_3$ reaction rates and exposure. Unreported alkenes are classified through comparison with reported alkenes in reaction rates of both $NO_3$ (kNO3) and $O_3$ (kO3) (Fig. S3). For example, the nocturnal consumption of 1-butene is over 96% through reaction with $O_3$ (de Gouw et al., 2017). The kO3 for 1-pentene is higher than the kO3 for 1-butene, but the kNO3 for 1-pentene is lower than the kNO3 for 1-butene. Therefore, the emitted 1-pentene concentrations are estimated using the $O_3$ exposure method. Briefly, styrene and 1,3-butadiene are determined using the $NO_3$ exposure method, while eight of the ten alkenes are determined using the $O_3$ exposure method.*

*During the daytime from 7:00 to 19:00 (Fig. S2), $NO_3$ is highly unstable and rapidly photolyzed. Therefore, VOCs consumption by its oxidation is negligible. Alkenes and styrene can react with both OH and $O_3$. For alkenes and styrene, the ratio of the product of the OH reaction rates (Carter, 2010) and the ambient OH concentration in the Chengdu Plain ($6.14 \times 10^6$ molecules $cm^{-3}$; (Yang et al., 2021) to the product of the $O_3$ reaction rates (Carter, 2010; Atkinson and Arey, 2003) and the ambient $O_3$ concentration (45.71 ppbv ) is 19.20. This indicates that OH predominantly consumes VOCs during the daytime. The emitted concentrations of NMHCs and MTBE are quantified during the daytime using the OH exposure method in Section 2.2.1. The OH exposure method is similar to the $NO_3$ and $O_3$ exposure methods,*

*Second, Brown et al. (2009b) calculated emitted isoprene concentrations during nighttime based on the steady-state $NO_3$ production from the reaction of $NO_2$ with $O_3$ and its consumption by isoprene. The mean ratios of measured $O_3$ to $NO_2$ concentrations during nighttime are 4.64 ppbv $ppbv^{-1}$ at Deyang, 1.42 ppbv $ppbv^{-1}$ at Chengdu, and 2.23 ppbv $ppbv^{-1}$ at Meishan. The reported method is not suitable for this study, because $O_3$ concentrations must be much larger than the $NO_2$ concentrations (Brown et al., 2009a).*

*Similar to styrene and 1,3-butadiene, emitted isoprene concentrations are determined using the NO₃ exposure method during nighttime. During the day, emitted isoprene concentrations are calculated using the OH exposure method and the ambient concentrations of MVK and MACR. Emitted MVK and MACR concentrations are calculated based on their measured concentrations and isoprene consumption.*

*Third, emitted OVOCs concentrations are determined using the photochemical age method during the daytime, due to its primary emissions, secondary production, and consumption by both OH and photon (hv). Among meteorological factors, temperature is the primary driver of $O_3$ production (Jacob and Winner, 2009), with the OH reaction rate showing small variations between 25°C and 35°C. For example, the reaction rate ratio for isoprene between these two temperatures is 0.96 (Saunders et al., 2003). Consequently, all VOCs reaction rate constants are adjusted for a temperature of 300 K.*

*2.2.1 NMHCs concentrations emitted by anthropogenic activities*

*For nighttime $NO_3$ consumption, the emitted concentrations of styrene and 1,3-butadiene are estimated using the $NO_3$ exposure method (de Gouw et al., 2017):*

$$[NO_3]\Delta t = \frac{1}{(kNO3_{benzene} - kNO3_{styrene})} \times \left[ln\left(\frac{[e\text{-}benzene]}{[e\text{-}styrene]}\right) - ln\left(\frac{[a\text{-}benzene]}{[a\text{-}styrene]}\right)\right] \quad (Eq.\ 1)$$

$$[e\text{-}alkene_j] = [a\text{-}alkene_j] \times exp(kNO3_{alkene_j}[NO_3]\Delta t) \quad (Eq.\ 2)$$

*where $[NO_3]$ and $\Delta t$, together referred to as $NO_3$ exposure ($[NO_3]\Delta t$), are the $NO_3$ concentrations and nocturnal reaction time, respectively. $kNO3_{benzene}$ and $kNO3_{styrene}$ are the reaction rate constants of benzene ($3.0 \times 10^{-17}$ cm³ molecule⁻¹ s⁻¹) and isoprene ($1.5 \times 10^{-13}$ cm³ molecule⁻¹ s⁻¹) with $NO_3$, respectively (Carter, 2010; Atkinson and Arey, 2003). [e-benzene]/[e-isoprene] is the emission ratio between benzene and isoprene. The estimated emission ratios are 1.0 ± 0.4ppbv ppbv⁻¹ at Deyang, 1.1 ± 0.5ppbv ppbv⁻¹ at Chengdu, and 2.7 ± 0.5ppbv ppbv⁻¹ at Meishan based on the measured data with a low degree of nocturnal consumption, respectively (Fig. S4). [a-benzene]/[a-styrene] is the hourly ambient concentration ratio between benzene and isoprene. $[a\text{-}alkenes_j]$ and $kNO3_{alkenej}$ refer to the ambient concentrations and $NO_3$ reaction rate constants (Fig. S3) of species j in styrene or 1,3-butadiene, respectively.*

*For nighttime $O_3$ consumption, the emitted concentrations of eight reactive alkenes are estimated using the $O_3$ exposure method (de Gouw et al., 2017):*

$$[O_3]\Delta t = \frac{1}{(kO3_{benzene} - kO3_{cis\text{-}2\text{-}butene})} \times \left[ln\left(\frac{[e\text{-}benzene]}{[e\text{-}cis\text{-}2\text{-}butene]}\right) - ln\left(\frac{[a\text{-}benzene]}{[a\text{-}cis\text{-}2\text{-}butene]}\right)\right] \quad (Eq.\ 3)$$

$$[e\text{-}alkene_j] = [a\text{-}alkene_j] \times exp(kO3_{alkene_j}[O_3]\Delta t) \quad (Eq.\ 4)$$

*where $[O_3]$ and $\Delta t$, together referred to as $O_3$ exposure ($[O_3]\Delta t$), are the $O_3$ concentrations and nocturnal reaction time. $kO3_{benzene}$ and $kO3_{cis\text{-}2\text{-}butene}$ are the reaction rate constants of benzene ($1.0 \times 10^{-20}$ cm³ molecule⁻¹ s⁻¹) and cis-2-butene ($1.3 \times 10^{-16}$ cm³ molecule⁻¹ s⁻¹) with $O_3$ (Carter, 2010; Atkinson and Arey, 2003). [e-benzene]/[e-cis-2-butene] is the emission ratios between benzene and cis-2-butene. The estimated emission*

ratios are 0.5 ± 0.3ppbv ppbv$^{-1}$ at Deyang, 4.5 ± 1.0ppbv ppbv$^{-1}$ at Chengdu, and 6.5 ± 1.0ppbv ppbv$^{-1}$ at Meishan based on measured data with a low degree of nocturnal consumption, respectively (Fig. S5). Similar to emission ratios of benzene to isoprene, emission ratios of benzene to cis-2-butene may remain consistent for each source. After mixing from different sources, the emission ratios obtained at different sampling sites may vary. [a-benzene]/[a-cis-2-butene] is the hourly ambient concentration ratio between benzene and cis-2-butene. [a-alkenes$_j$] and kO3$_{alkenej}$ refer to the ambient concentrations and O$_3$ reaction rate constants (Fig. S3) of the species j in alkenes, respectively. Cis-2-butene is replaced with trans-2-butene at Chengdu due to the unavailability of cis-2-butene data.

For daytime OH consumption, the emitted concentrations of each NMHC [e-NMHC], including MTBE, are estimated using the OH exposure method (Ma et al., 2022; Shao et al., 2011; de Gouw et al., 2005; Roberts et al., 1984):

$$[OH]\Delta t = \frac{1}{(kOH_{ethylbenzene} - kOH_{m,p-xylenes})} \times \left[ln\left(\frac{[e\text{-}ethylbenzene]}{[e\text{-}m,p\text{-}xylenes]}\right) - ln\left(\frac{[a\text{-}ethylbenzene]}{[a\text{-}m,p\text{-}xylenes]}\right)\right] \quad (Eq.\ 5)$$

$$[e\text{-}NHMC_j] = [a\text{-}NHMC_j] \times exp(kOH_{NMHCj}[OH]\Delta t) \quad (Eq.\ 6)$$

where [OH] and Δt, together referred to as OH exposure ([OH]Δt), are the OH concentrations and reaction time, respectively. [e-ethylbenzene]/[e-m,p-xylenes] is the emission ratio between ethylbenzene and m,p-xylenes (Fig. S6). The major source of ethylbenzene and m,p-xylenes in the Chengdu Plain is solvent use (Wu and Xie, 2017). There is a strong linear correlation between ethylbenzene and m,p-xylenes (R² = 0.96). kOH$_{ethylbenzene}$ and kOH$_{m,p-xylenes}$ are the reaction rate constants of ethylbenzene (7.0 × 10$^{-12}$ cm$^3$ molecule$^{-1}$ s$^{-1}$) and m,p-xylenes (1.9 × 10$^{-11}$ cm$^3$ molecule$^{-1}$ s$^{-1}$) with OH, respectively (Carter, 2010). [a-NMHC$_j$] and kOH$_{NMHCj}$ denote the hourly ambient concentrations and OH reaction rate constants (Fig. S7) of the species j in NMHCs, respectively.

*2.2.2 Isoprene concentrations emitted by biogenic sources*

For nighttime NO$_3$ consumption, the emitted concentrations of isoprene are estimated using the NO$_3$ exposure method (de Gouw et al., 2017):

$$[NO_3]\Delta t = \frac{1}{(kNO3_{MVK} - kNO3_{isoprene})} \times \left[ln\left(\frac{[e\text{-}MVK]}{[e\text{-}isoprene]}\right) - ln\left(\frac{[a\text{-}MVK]}{[a\text{-}isoprene]}\right)\right] \quad (Eq.\ 7)$$

$$[e\text{-}isoprene] = [a\text{-}isoprene] \times exp(kNO3_{isoprene}[NO_3]\Delta t) \quad (Eq.\ 8)$$

where [NO$_3$] and Δt, together referred to as NO$_3$ exposure ([NO$_3$]Δt), are the concentrations of NO$_3$ and nocturnal reaction time, respectively. kNO3$_{MVK}$ and kNO3$_{isoprene}$ are the reaction rate constants of MVK (5.4 × 10$^{-18}$ cm$^3$ molecule$^{-1}$ s$^{-1}$ with O$_3$) and isoprene (6.8 × 10$^{-13}$ cm$^3$ molecule$^{-1}$ s$^{-1}$) with NO$_3$, respectively (Carter, 2010; Atkinson and Arey, 2003). The kNO3$_{MVK}$ value is very small. Due to the unavailability of the kNO3$_{MVK}$ value, kO3$_{MVK}$ is used as a substitute. [e-MVK]/[e-isoprene] is the emission ratio between MVK and isoprene. Although MVK and isoprene emissions are low at night, many field studies have demonstrated that they can accumulate in the early nighttime from

20:00 to 21:00 (Wennberg et al., 2018). Therefore, the measured MVK and isoprene concentrations in the early nighttime are the "emitted" concentrations for nighttime $NO_3$ consumption. The estimated emission ratios are $0.5 \pm 0.2$ ppbv ppbv$^{-1}$ at Deyang, $0.1 \pm 0.1$ ppbv ppbv$^{-1}$ at Chengdu, and $0.1 \pm 0.1$ ppbv ppbv$^{-1}$ at Meishan from measured data with a low degree of nocturnal consumption, respectively (Fig. S8). There are no significant differences in the estimated emission ratios between early and late nighttime. Therefore, nighttime low MVk and isoprene emissions may not influence this calculation method. The emission ratios are directly linked to emission sources. After mixing from different sources, the emission ratios obtained at different sampling sites may vary. Although MVK and isoprene may originate from different sources during the nighttime, both anthropogenic and biogenic activities in the Chengdu Plain are relatively stable at nighttime based on both our unpublished results and the reported findings of a study using positive matrix factorization (Zheng et al., 2023; Kong et al., 2023; Xiong et al., 2021). Furthermore, as surrogates for traffic flows, the traffic congestion indices during the nighttime in Chengdu remain relatively stable (https://jiaotong.baidu.com/congestion/city/urbanrealtime). Therefore, their emission ratios may remain consistent. [a-MVK]/[a-isoprene] is the hourly ambient concentration ratio between MVK and isoprene. MACR is not used as a stable biogenic tracer due to its relatively high $NO_3$ reaction rate ($3.5 \times 10^{-15}$ cm$^3$ molecule$^{-1}$ s$^{-1}$) compared to MVK.

For daytime OH consumption, the emitted isoprene concentrations are estimated using the OH exposure method in Eqs. (9-12) (Paulot et al., 2009; Stroud et al., 2001).

$$[OH]\Delta t_{MVK} = \frac{1}{(kOH_{isoprene} - kOH_{MVK})} \times ln\left(1 - \frac{[a\text{-}MVK]}{[a\text{-}isoprene]} \times \frac{kOH_{MVK} - kOH_{isoprene}}{0.32 \times kOH_{isoprene}}\right) \quad (Eq.\ 9)$$

$$[OH]\Delta t_{MACR} = \frac{1}{(kOH_{isoprene} - kOH_{MACR})} \times ln\left(1 - \frac{[a\text{-}MACR]}{[a\text{-}isoprene]} \times \frac{kOH_{MACR} - kOH_{isoprene}}{0.23 \times kOH_{isoprene}}\right) (Eq.\ 10)$$

$$[OH]\Delta t_{isoprene} = ([OH]\Delta t_{MVK} + [OH]\Delta t_{MACR}) / 2 \quad (Eq.\ 11)$$

$$[e\text{-}isoprene] = [a\text{-}isoprene] \times exp(kOH_{isoprene}[OH]\Delta t_{isoprene}) \quad (Eq.\ 12)$$

where $kOH_{isoprene}$, $kOH_{MVK}$, and $kOH_{MACR}$ are the reaction rate constants of isoprene ($1 \times 10^{-10}$ cm$^3$ molecule$^{-1}$ s$^{-1}$), MVK ($2.0 \times 10^{-11}$ cm$^3$ molecule$^{-1}$ s$^{-1}$), and MACR ($2.8 \times 10^{-11}$ cm$^3$ molecule$^{-1}$ s$^{-1}$) with OH (Carter, 2010; Atkinson and Arey, 2003). [a-isoprene], [a-MVK], and [a-MACR] refer to the ambient concentrations of isoprene, MVK, and MACR, respectively.

For daytime OH and hv consumption, the emitted concentrations of MVK and MACR ([e-OVOC]) are estimated based on isoprene consumption:

$$[c\text{-}isoprene] = [e\text{-}isoprene] - [a\text{-}isoprene] \quad (Eq.\ 13)$$

$$[s\text{-}OVOC_j] = p \times [c\text{-}isoprene] \quad (Eq.\ 14)$$

$$[e\text{-}OVOC_j] = [a\text{-}OVOC_j] - [s\text{-}OVOC_j] + [c\text{-}isoprene] \times \frac{([a\text{-}OVOC_j] - [s\text{-}OVOC_j]) \times kOH^*_{OVOC_j}}{[a\text{-}isoprene] \times kOH_{isoprene}}$$

$$(Eq.\ 15)$$

where [c-isoprene] indicates consumed concentrations of isoprene, which are equivalent to its emitted concentrations ([e-isoprene]) calculated from Eq. (12) minus the ambient concentrations ([a-isoprene]). [s-OVOC$_j$] represents the secondary concentrations of species j in MVK or MACR produced from isoprene oxidation. The p values in Eq. (14) represent the molecular production from one molecular unit of isoprene consumption, with values of 0.32 for MVK and 0.23 for MACR, respectively (Paulot et al., 2009). Because the kOH$_{MVK}$ and kOH$_{MACR}$ values are 3.5 times and 5 times lower than those of kOH$_{isoprene}$ (Fig. S7), respectively, the ambient concentrations ([a-OVOC$_j$]) are assumed to be instantaneous total concentrations in Eq. (15). Therefore, [e-OVOC$_j$] is approximately equal to [a-OVOC$_j$] minus [s-OVOC$_j$] plus the corresponding photochemical consumption, which is calculated using [c-isoprene] and the reaction rates. The photochemical consumption of OVOCs includes both photolysis and reaction with OH. Therefore, the total OVOC$_j$ loss rates (kOH* OVOC$_j$) are estimated based on the photolysis rate (J$_{OVOCj}$) and the loss rate with OH ([OH]kOH$_{OVOCj}$). The ratio of the J$_{NO2}$ to OH concentrations in the Sichuan Basin is similar to that reported in the Los Angeles Basin (Yang et al., 2021; de Gouw et al., 2018); therefore the J$_{OVOCj}$ and [OH]kOH$_{OVOCj}$ may be comparable (Tan et al., 2018; de Gouw et al., 2018). Accordingly, we assume that the ratios (0.6 for MVK and MACR) of J$_{OVOCj}$ to [OH]kOH$_{OVOCj}$ established in the Los Angeles Basin (Fig. S9; (de Gouw et al., 2018) are applicable for estimating kOH* OVOC$_j$ = (1 + 0.6) × kOH$_{OVOCj}$ in the Sichuan Basin. kOH$_{isoprene}$ is the OH reaction rate constant of isoprene.

**2.2.3 OVOCs concentrations emitted by both anthropogenic and biogenic sources**

To differentiate secondary production and consumption, we estimate the emitted concentrations of ten of the 13 OVOCs ([e-OVOC]) during the daytime using the photochemical age method in Eqs. (16-17) (Wu et al., 2020; de Gouw et al., 2018; de Gouw et al., 2005). MTBE is excluded, with details provided in Section 2.2.1. The estimation methods for MVK and MACR are described in Section 2.2.2.

$$[a\text{-}OVOC_j] = ER_{OVOC_j} \times [a\text{-}benzene] \times exp(-(kOH^*_{OVOC_j} - kOH_{benzene})[OH]\Delta t) +$$

$$ER_{HC} \times [a\text{-}benzene] \times \frac{kOH_{HC}}{kOH^*_{OVOC_j} - kOH_{HC}} \times \frac{exp(-kOH_{HC}[OH]\Delta t) - exp(-kOH^*_{OVOC_j}[OH]\Delta t)}{exp(-kOH_{benzene}[OH]\Delta t))} +$$

$$ER_{biogenic} \times [e\text{-}isoprene] \qquad\qquad\qquad (Eq.\ 16)$$

$$[e\text{-}OVOC_j] = ER_{OVOC_j} \times [a\text{-}benzene] + ER_{biogenic} \times [e\text{-}isoprene] \qquad (Eq.\ 17)$$

where the measured concentrations of species j in OVOCs ([a-OVOC$_j$]) are equivalent to the sum of primary anthropogenic contributions, secondary anthropogenic contributions, and biogenic contributions, as represented sequentially in Eq. (16). Benzene is selected as the tracer of anthropogenic primary sources due to the dominance of combustion and industrial VOCs emissions in the Sichuan Basin (Wu and Xie, 2017) and its relatively low OH reaction rate. ER$_{OVOCj}$ and ER$_{HC}$ are the emission ratios of species j in OVOCs and hydrocarbons to benzene, respectively. We assume that the ratios (R) for

*$J_{OVOCj}$ to [OH]kOH_{OVOCj} established in the Los Angeles Basin (Fig. S9; (de Gouw et al., 2018) are applicable for estimating $kOH^* OVOCj = (1 + R) \times kOH_{OVOCj}$ in the Sichuan Basin. OH exposure is calculated using Eq. (5). $ER_{biogenic}$ is the emission ratio between OVOCs and isoprene from biogenic sources. [e-isoprene] is estimated by Eqs. (9-12). The $ER_{OVOCj}$, $ER_{HC}$, $kOH_{HC}$, and $ER_{biogenic}$ values are determined using the nonlinear least-squares fit.'*

3.  English language comment: The paper uses past tense frequently where present tense make more sense for English writing. If possible, suggest proofreading by a native English speaker. As one example, the first line of the manuscript (line 24) is better write as "Volatile organic compounds are key species in ozone formation".

**Response:**

We have paid for native English speakers to polish our writing. We have also reviewed the differences between original and polished manuscripts once again. The following paragraph is the explanation after being polished.

*'The English in this document has been checked by at least two professional editors, both native speakers of English.'*

4.  Line 28: The terminology of "initial VOCs" to indicate VOC emissions is somewhat confusing. Would suggest a different label for these compounds to make this clear, as in "emitted VOCs".

**Response:**

The term "initial VOCs" has been replaced with "emitted VOCs (e-VOCs)." Please review the text, figures, and tables accordingly.

5.  Line 81: replace "moving" with "removing" and "particulate matters" with "particulate matter"

**Response:** Done (please check line 83 in Materials and Methods).

6.  Line 97-98: The "$NO_3$ or $O_3$ exposure method" is not defined. How is this used to estimate emissions?

**Response:**

We define the $NO_3$ or $O_3$ exposure method and explain the theoretical basis for estimating emitted concentrations (please check lines 109-111 in Materials and Methods). The details of calculation methods are given in Eqs. 1-4 and 7-8.

*'The $NO_3$ or $O_3$ exposure method indicates that the concentration ratios of a stable tracer species to a reactive tracer species would increase with both $NO_3$ or $O_3$ concentrations and reaction time after emissions.'*

7. Line 117 and equation 1: Benzene and isoprene are not co-emitted species, so no relationship or emission ratio would be expected between these species. Furthermore, these species have very different diel emissions profiles, with isoprene not emitted at night but strongly peaked during daytime. The indicated dependence of the benzene to isoprene ratio is therefore not likely to be applicable to this pair. A different approach would be required to relate the concentrations of these species that lack a common emission source. Other pairs, such as benzene – styrene, or benzene – butadiene that have common anthropogenic sources should be amenable to this method.

**Response:**

We choose benzene and styrene for the calculation of NMHCs concentrations emitted by anthropogenic activities. We revise the sentences (please check lines 138-149 in Materials and Methods).

*'For nighttime $NO_3$ consumption, the emitted concentrations of styrene and 1,3-butadiene are estimated using the $NO_3$ exposure method (de Gouw et al., 2017):*

$$[NO_3]\Delta t = \frac{1}{(kNO3_{benzene} - kNO3_{styrene})} \times \left[ ln\left(\frac{[e\text{-}benzene]}{[e\text{-}styrene]}\right) - ln\left(\frac{[a\text{-}benzene]}{[a\text{-}styrene]}\right) \right] \qquad (Eq.\ 1)$$

$$[e\text{-}alkene_j] = [a\text{-}alkene_j] \times exp(kNO3_{alkene_j}[NO_3]\Delta t) \qquad (Eq.\ 2)$$

*where $[NO_3]$ and $\Delta t$, together referred to as $NO_3$ exposure ($[NO_3]\Delta t$), are the $NO_3$ concentrations and nocturnal reaction time, respectively. $kNO3_{benzene}$ and $kNO3_{styrene}$ are the reaction rate constants of benzene $(3.0 \times 10^{-17}\ cm^3\ molecule^{-1}\ s^{-1})$ and isoprene $(1.5 \times 10^{-13}\ cm^3\ molecule^{-1}\ s^{-1})$ with $NO_3$, respectively (Carter, 2010; Atkinson and Arey, 2003). [e-benzene]/[e-isoprene] is the emission ratio between benzene and isoprene. The estimated emission ratios are $1.0 \pm 0.4ppbv\ ppbv^{-1}$ at Deyang, $1.1 \pm 0.5ppbv\ ppbv^{-1}$ at Chengdu, and $2.7 \pm 0.8ppbv\ ppbv^{-1}$ at Meishan based on the measured data with a low degree of nocturnal consumption, respectively (Fig. S4). [a-benzene]/[a-styrene] is the hourly ambient concentration ratio between benzene and isoprene. $[a\text{-}alkenes_j]$ and $kNO3_{alkenej}$ refer to the ambient concentrations and $NO_3$ reaction rate constants (Fig. S3) of species j in styrene or 1,3-butadiene, respectively.'*

8. Line 271 and figure S10: It does not necessarily make sense to calculate an OFP during nighttime hours.

**Response:**

We compare potential differences in the OFP between ambient and emitted VOCs concentrations even during the nighttime. If not suitable, we also emphasize the distinction between daytime and nighttime in Figures S10 and S11. We revise the sentences (please check lines 316-350 in Results and discussion).

*'Although the OFP is primarily associated with daytime $O_3$ production, nighttime OFP is also calculated to compare the potential differences in the contributions of VOCs and emitted VOCs to the OFP (Figures 10 and 11). The temporal variation trends in the*

OFP values are relatively consistent at the three sites based on emitted and ambient TVOCs concentrations (Figure S10ab). The OFP based on emitted and ambient TVOCs concentrations ranges from 65.60 μg m$^{-3}$ to 1476.28 μg m$^{-3}$ and from 55.86 μg m$^{-3}$ to 827.30 μg m$^{-3}$, respectively, at Deyang. Among VOCs chemical groups, alkenes exhibit the greatest variations in OFP, with ranges of 15.12 μg m$^{-3}$ to 1081.41 μg m$^{-3}$ and 7.87 μg m$^{-3}$ to 282.78 μg m$^{-3}$, respectively. The highest OFP based on the emitted TVOCs concentrations was found at 16:00 on 11 August 2019, primarily due to the 1081.41 μg m$^{-3}$ contribution from alkenes. At this time, the OFP based on the ambient TVOCs concentrations is only 327.41 μg m$^{-3}$. The ratio of the OFP between the emitted and ambient TVOCs concentrations exceeds 4.5. This discrepancy is mainly because of the large consumption of alkenes at that time. If the consumption of alkenes is not considered, their contributions to the OFP would be greatly underestimated.

The OFP based on emitted and ambient TVOCs concentrations ranges from 47.47 μg m$^{-3}$ to 1143.74 μg m$^{-3}$ and from 39.21 μg m$^{-3}$ to 819.61 μg m$^{-3}$, respectively at Chengdu (Figure S10cd). Alkenes exhibit the greatest variations in OFP among VOCs chemical groups, with ranges from 12.34 μg m$^{-3}$ to 780.85 μg m$^{-3}$ and from 4.94 μg m$^{-3}$ to 336.58 μg m$^{-3}$, respectively. The highest OFP based on the emitted TVOCs concentrations was observed at 12:00 on 30 August 2019, which is primarily due to the contributions of 500.64 μg m$^{-3}$ from alkenes and 494.89 μg m$^{-3}$ from aromatics. At this time, the OFP based on the ambient TVOCs concentrations is 819.61 μg m$^{-3}$. The OFP based on the emitted and ambient TVOCs concentrations are similar, indicating that photochemical consumption is relatively low at Chengdu compared to Deyang. Furthermore, the OFP of the ambient OVOCs concentrations is 171.94 μg m$^{-3}$, which is higher than the OFP of emitted concentrations at 88.40 μg m$^{-3}$. Compared to the other two sites, emissions from solvent use are higher around Chengdu. This leads to higher emissions of aromatics, which significantly contributes to the OFP.

The OFP based on emitted and ambient TVOCs concentrations ranges from 59.30 μg m$^{-3}$ to 1351.58 μg m$^{-3}$ and from 48.43 μg m$^{-3}$ to 1077.27 μg m$^{-3}$, respectively at Meishan (Figure S10ef). Alkenes display the greatest variations in OFP, ranging from 13.98 μg m$^{-3}$ to 1133.53 μg m$^{-3}$ and from 7.62 μg m$^{-3}$ to 864.52 μg m$^{-3}$, respectively. The highest OFP of the emitted TVOCs concentrations was found at 18:00 on 17 August 2019, due to the major contribution of 1133.53 μg m$^{-3}$ from alkenes. Because of low photochemical consumption and secondary OVOCs formation, the difference in the OFP between emitted and ambient TVOCs concentrations was relatively low, at about 25% at this time. Compared to the other two sites, there are more biogenic isoprene emissions around Meishan, which contributes to the OFP of alkenes.

The highest hourly OFP values of emitted TVOCs concentrations are 391.07 μg m$^{-3}$ at 12:00 at Deyang, 432.12 μg m$^{-3}$ at 12:00 at Chengdu, and 403.80 μg m$^{-3}$ at 18:00 at Meishan, respectively (Figure S11). After considering the nighttime alkene consumption, the emitted alkene concentrations are close to those during the day at Deyang. Moreover,

*the MIR values of alkenes are generally higher across these VOCs chemical groups (Carter, 2010). These may explain why the total OFP at night is similar to that during the day at Deyang. The diurnal variations in the OFP at Chengdu are consistent with those of sunlight intensity. The OFP at Meishan is primarily driven by isoprene. The surrounding bamboo forest acts as a source of biogenic emissions. The accumulation of isoprene causes the highest concentrations, and thus OFP, occurring at 18:00.'*

9.  Line 309: Isoprene rather than Ioprene

**Response:** Done (please check line 358 in Results and discussion).

**Reviewer #3**

1.  Having hourly concentrations of 99 VOCs and OVOCs at three different field sites in the Sichuan Basin, China, Zheng and Xie calculate the ozone formation potential (OFP) for these species using the maximum incremental reactivity (MIR) method. However, unlike previous studies that have calculated OFPs using the observed/measured concentration of the VOC or OVOC, the authors correct these concentrations to estimate what the concentrations of these species would have been before being consumed or produced in the atmosphere (something they refer to as "initial" concentrations). The authors show that using the "initial" concentration of a species can impact the relative importance of that species in forming ozone.

    The paper would be appropriate for inclusion in ACP after the following comments are addressed:

**Response:** Thank you for reviewing our manuscript.

2.  Eqs. 1, 3, and 5, there should be a brief explanation for each on why benzene and isoprene, benzene and cis-2-butene, and ethylbenzene and m,p-xylenes are paired to calculate the NO3, O3, and OH exposures, respectively. For example, in the NO3 exposure, benzene and isoprene are paired, but they have completely different emission sources and do not seem well correlated in Figure S4, so explaining why they are paired would be beneficial to explain for the reader.

**Response:**

We have revised the overall classification method in Section 2.2. The emitted VOC concentrations are classified into three categories: anthropogenic sources in Section 2.2.1, biogenic sources in Section 2.2.2, and a combination of both in Section 2.2.3. VOCs species pairs with common sources were selected to calculate the NO₃, O₃, and OH exposure in Sections 2.2.1 and 2.2.2. We revise the sentences (please check lines 97-131 and 138-207 in Materials and methods). We also update the comparison between emitted

and ambient VOC concentrations, as well as their corresponding OFP values in Results and discussion.

**'2.2 Calculations of emitted VOCs concentrations**

*Source classification is crucial for calculating emitted VOCs concentrations. NMHCs (except isoprene) and MTBE are generally emitted from anthropogenic activities. Isoprene is typically emitted from biogenic sources and oxidized into methyl vinyl ketone (MVK) and methacrolein (MACR). In addition to secondary production, ten of the 13 OVOCs (except MTBE, MVK, and MACR) are also emitted from both anthropogenic and biogenic sources (Zou et al., 2024; Lyu et al., 2024; Li et al., 2023; Wu et al., 2020). The emitted VOC concentrations are classified into three categories: anthropogenic sources in Section 2.2.1, biogenic sources in Section 2.2.2, and a combination of both in Section 2.2.3.*

*First, the major atmospheric oxidants for the consumption of emitted VOCs are $NO_3$, $O_3$, and OH. Due to the absence of sunlight, OH concentrations and photolysis rates are very low from 20:00 to 06:00 (Fig. S2). Either $NO_3$ or $O_3$ are the primary oxidants for the consumption of emitted alkene and styrene during the nighttime. During the local nighttime, emitted alkene and styrene concentrations are estimated through the $NO_3$ or $O_3$ exposure methods based on the relative loss rates of reported species between $NO_3$ and $O_3$ in the Los Angeles Basin (de Gouw et al., 2017). Other VOCs are excluded from the analysis due to their slow reaction rates with $NO_3$ and $O_3$ during nighttime. The $NO_3$ or $O_3$ exposure method indicates that the concentration ratios of a stable tracer species to a reactive tracer species would increase with both $NO_3$ or $O_3$ concentrations and reaction time after emissions. Emitted concentrations are calculated based on $NO_3$ or $O_3$ reaction rates and exposure. Unreported alkenes are classified through comparison with reported alkenes in reaction rates of both $NO_3$ (kNO3) and $O_3$ (kO3) (Fig. S3). For example, the nocturnal consumption of 1-butene is over 96% through reaction with $O_3$ (de Gouw et al., 2017). The kO3 for 1-pentene is higher than the kO3 for 1-butene, but the kNO3 for 1-pentene is lower than the kNO3 for 1-butene. Therefore, the emitted 1-pentene concentrations are estimated using the $O_3$ exposure method. Briefly, styrene and 1,3-butadiene are determined using the $NO_3$ exposure method, while eight of the ten alkenes are determined using the $O_3$ exposure method.*

*During the daytime from 7:00 to 19:00 (Fig. S2), $NO_3$ is highly unstable and rapidly photolyzed. Therefore, VOCs consumption by its oxidation is negligible. Alkenes and styrene can react with both OH and $O_3$. For alkenes and styrene, the ratio of the product of the OH reaction rates (Carter, 2010) and the ambient OH concentration in the Chengdu Plain ($6.14 \times 10^6$ molecules $cm^{-3}$; (Yang et al., 2021) to the product of the $O_3$ reaction rates (Carter, 2010; Atkinson and Arey, 2003) and the ambient $O_3$ concentration (45.71 ppbv ) is 19.20. This indicates that OH predominantly consumes VOCs during the daytime. The emitted concentrations of NMHCs and MTBE are quantified during the daytime using*

*the OH exposure method in Section 2.2.1. The OH exposure method is similar to the $NO_3$ and $O_3$ exposure methods,*

*Second, Brown et al. (2009b) calculated emitted isoprene concentrations during nighttime based on the steady-state $NO_3$ production from the reaction of $NO_2$ with $O_3$ and its consumption by isoprene. The mean ratios of measured $O_3$ to $NO_2$ concentrations during nighttime are 4.64 ppbv ppbv$^{-1}$ at Deyang, 1.42 ppbv ppbv$^{-1}$ at Chengdu, and 2.23 ppbv ppbv$^{-1}$ at Meishan. The reported method is not suitable for this study, because $O_3$ concentrations must be much larger than the $NO_2$ concentrations (Brown et al., 2009a). Similar to styrene and 1,3-butadiene, emitted isoprene concentrations are determined using the $NO_3$ exposure method during nighttime. During the day, emitted isoprene concentrations are calculated using the OH exposure method and the ambient concentrations of MVK and MACR. Emitted MVK and MACR concentrations are calculated based on their measured concentrations and isoprene consumption.'*

*'2.2.1 NMHCs concentrations emitted by anthropogenic activities*

*For nighttime $NO_3$ consumption, the emitted concentrations of styrene and 1,3-butadiene are estimated using the $NO_3$ exposure method (de Gouw et al., 2017):*

$$[NO_3]\Delta t = \frac{1}{(kNO3_{benzene} - kNO3_{styrene})} \times \left[ ln\left(\frac{[e\text{-}benzene]}{[e\text{-}styrene]}\right) - ln\left(\frac{[a\text{-}benzene]}{[a\text{-}styrene]}\right)\right] \quad (Eq.\ 1)$$

$$[e\text{-}alkene_j] = [a\text{-}alkene_j] \times exp(kNO3_{alkene_j}[NO_3]\Delta t) \quad (Eq.\ 2)$$

*where $[NO_3]$ and $\Delta t$, together referred to as $NO_3$ exposure ($[NO_3]\Delta t$), are the $NO_3$ concentrations and nocturnal reaction time, respectively. $kNO3_{benzene}$ and $kNO3_{styrene}$ are the reaction rate constants of benzene ($3.0 \times 10^{-17}$ cm$^3$ molecule$^{-1}$ s$^{-1}$) and isoprene ($1.5 \times 10^{-13}$ cm$^3$ molecule$^{-1}$ s$^{-1}$) with $NO_3$, respectively (Carter, 2010; Atkinson and Arey, 2003). $[e\text{-}benzene]/[e\text{-}isoprene]$ is the emission ratio between benzene and isoprene. The estimated emission ratios are $1.0 \pm 0.4$ppbv ppbv$^{-1}$ at Deyang, $1.1 \pm 0.5$ppbv ppbv$^{-1}$ at Chengdu, and $2.7 \pm 0.5$ppbv ppbv$^{-1}$ at Meishan based on the measured data with a low degree of nocturnal consumption, respectively (Fig. S4). $[a\text{-}benzene]/[a\text{-}styrene]$ is the hourly ambient concentration ratio between benzene and isoprene. $[a\text{-}alkenes_j]$ and $kNO3_{alkenej}$ refer to the ambient concentrations and $NO_3$ reaction rate constants (Fig. S3) of species j in styrene or 1,3-butadiene, respectively.*

*For nighttime $O_3$ consumption, the emitted concentrations of eight reactive alkenes are estimated using the $O_3$ exposure method (de Gouw et al., 2017):*

$$[O_3]\Delta t = \frac{1}{(kO3_{benzene} - kO3_{cis\text{-}2\text{-}butene})} \times \left[ ln\left(\frac{[e\text{-}benzene]}{[e\text{-}cis\text{-}2\text{-}butene]}\right) - ln\left(\frac{[a\text{-}benzene]}{[a\text{-}cis\text{-}2\text{-}butene]}\right)\right] \quad (Eq.\ 3)$$

$$[e\text{-}alkene_j] = [a\text{-}alkene_j] \times exp(kO3_{alkene_j}[O_3]\Delta t) \quad (Eq.\ 4)$$

*where $[O_3]$ and $\Delta t$, together referred to as $O_3$ exposure ($[O_3]\Delta t$), are the $O_3$ concentrations and nocturnal reaction time. $kO3_{benzene}$ and $kO3_{cis\text{-}2\text{-}butene}$ are the reaction rate constants of benzene ($1.0 \times 10^{-20}$ cm$^3$ molecule$^{-1}$ s$^{-1}$) and cis-2-butene ($1.3 \times 10^{-16}$ cm$^3$ molecule$^{-1}$ s$^{-1}$) with $O_3$ (Carter, 2010; Atkinson and Arey, 2003). $[e\text{-}benzene]/[e\text{-}cis\text{-}2\text{-}*

butene] is the emission ratios between benzene and cis-2-butene. The estimated emission ratios are $0.5 \pm 0.3$ppbv ppbv$^{-1}$ at Deyang, $4.5 \pm 1.0$ppbv ppbv$^{-1}$ at Chengdu, and $6.5 \pm 1.0$ppbv ppbv$^{-1}$ at Meishan based on measured data with a low degree of nocturnal consumption, respectively (Fig. S5). Similar to emission ratios of benzene to isoprene, emission ratios of benzene to cis-2-butene may remain consistent for each source. After mixing from different sources, the emission ratios obtained at different sampling sites may vary. [a-benzene]/[a-cis-2-butene] is the hourly ambient concentration ratio between benzene and cis-2-butene. [a-alkenes$_j$] and kO3$_{alkenej}$ refer to the ambient concentrations and O$_3$ reaction rate constants (Fig. S3) of the species j in alkenes, respectively. Cis-2-butene is replaced with trans-2-butene at Chengdu due to the unavailability of cis-2-butene data.

For daytime OH consumption, the emitted concentrations of each NMHC [e-NMHC], including MTBE, are estimated using the OH exposure method (Ma et al., 2022; Shao et al., 2011; de Gouw et al., 2005; Roberts et al., 1984):

$$[OH]\Delta t = \frac{1}{(kOH_{ethylbenzene} - kOH_{m,p-xylenes})} \times \left[ ln\left(\frac{[e\text{-}ethylbenzene]}{[e\text{-}m,p\text{-}xylenes]}\right) - ln\left(\frac{[a\text{-}ethylbenzene]}{[a\text{-}m,p\text{-}xylenes]}\right) \right] \quad (Eq.\ 5)$$

$$[e\text{-}NHMC_j] = [a\text{-}NHMC_j] \times exp(kOH_{NMHC_j}[OH]\Delta t) \quad\quad (Eq.\ 6)$$

where [OH] and $\Delta t$, together referred to as OH exposure ([OH]$\Delta t$), are the OH concentrations and reaction time, respectively. [e-ethylbenzene]/[e-m,p-xylenes] is the emission ratio between ethylbenzene and m,p-xylenes (Fig. S6). The major source of ethylbenzene and m,p-xylenes in the Chengdu Plain is solvent use (Wu and Xie, 2017). There is a strong linear correlation between ethylbenzene and m,p-xylenes ($R^2 = 0.96$). kOH$_{ethylbenzene}$ and kOH$_{m,p-xylenes}$ are the reaction rate constants of ethylbenzene ($7.0 \times 10^{-12}$ cm$^3$ molecule$^{-1}$ s$^{-1}$) and m,p-xylenes ($1.9 \times 10^{-11}$ cm$^3$ molecule$^{-1}$ s$^{-1}$) with OH, respectively (Carter, 2010). [a-NMHC$_j$] and kOH$_{NMHCj}$ denote the hourly ambient concentrations and OH reaction rate constants (Fig. S7) of the species j in NMHCs, respectively.

*2.2.2 Isoprene concentrations emitted by biogenic sources*

For nighttime NO$_3$ consumption, the emitted concentrations of isoprene are estimated using the NO$_3$ exposure method (de Gouw et al., 2017):

$$[NO_3]\Delta t = \frac{1}{(kNO3_{MVK} - kNO3_{isoprene})} \times \left[ ln\left(\frac{[e\text{-}MVK]}{[e\text{-}isoprene]}\right) - ln\left(\frac{[a\text{-}MVK]}{[a\text{-}isoprene]}\right) \right] \quad\quad (Eq.\ 7)$$

$$[e\text{-}isoprene] = [a\text{-}isoprene] \times exp(kNO3_{isoprene}[NO_3]\Delta t) \quad\quad (Eq.\ 8)$$

where [NO$_3$] and $\Delta t$, together referred to as NO$_3$ exposure ([NO$_3$]$\Delta t$), are the concentrations of NO$_3$ and nocturnal reaction time, respectively. kNO3$_{MVK}$ and kNO3$_{isoprene}$ are the reaction rate constants of MVK ($5.4 \times 10^{-18}$ cm$^3$ molecule$^{-1}$ s$^{-1}$ with O$_3$) and isoprene ($6.8 \times 10^{-13}$ cm$^3$ molecule$^{-1}$ s$^{-1}$) with NO$_3$, respectively (Carter, 2010; Atkinson and Arey, 2003). The kNO3$_{MVK}$ value is very small. Due to the unavailability of the kNO3$_{MVK}$ value, kO3$_{MVK}$ is used as a substitute. [e-MVK]/[e-isoprene] is the emission ratio between MVK and isoprene. Although MVK and isoprene emissions are low at night,

*many field studies have demonstrated that they can accumulate in the early nighttime from 20:00 to 21:00 (Wennberg et al., 2018). Therefore, the measured MVK and isoprene concentrations in the early nighttime are the "emitted" concentrations for nighttime NO₃ consumption. The estimated emission ratios are 0.5 ± 0.2ppbv ppbv⁻¹ at Deyang, 0.1 ± 0.1ppbv ppbv⁻¹ at Chengdu, and 0.1 ± 0.1ppbv ppbv⁻¹ at Meishan from measured data with a low degree of nocturnal consumption, respectively (Fig. S8). There are no significant differences in the estimated emission ratios between early and late nighttime. Therefore, nighttime low MVk and isoprene emissions may not influence this calculation method. The emission ratios are directly linked to emission sources. After mixing from different sources, the emission ratios obtained at different sampling sites may vary. Although MVK and isoprene may originate from different sources during the nighttime, both anthropogenic and biogenic activities in the Chengdu Plain are relatively stable at nighttime based on both our unpublished results and the reported findings of a study using positive matrix factorization (Zheng et al., 2023; Kong et al., 2023; Xiong et al., 2021). Furthermore, as surrogates for traffic flows, the traffic congestion indices during the nighttime in Chengdu remain relatively stable (https://jiaotong.baidu.com/congestion/city/urbanrealtime). Therefore, their emission ratios may remain consistent. [a-MVK]/[a-isoprene] is the hourly ambient concentration ratio between MVK and isoprene. MACR is not used as a stable biogenic tracer due to its relatively high NO₃ reaction rate (3.5 × 10⁻¹⁵ cm³ molecule⁻¹ s⁻¹) compared to MVK.*

*For daytime OH consumption, the emitted isoprene concentrations are estimated using the OH exposure method in Eqs. (9-12) (Paulot et al., 2009; Stroud et al., 2001).*

$$[OH]\Delta t_{MVK} = \frac{1}{(kOH_{isoprene} - kOH_{MVK})} \times ln\left(1 - \frac{[a\text{-}MVK]}{[a\text{-}isoprene]} \times \frac{kOH_{MVK} - kOH_{isoprene}}{0.32 \times kOH_{isoprene}}\right) \quad (Eq.\ 9)$$

$$[OH]\Delta t_{MACR} = \frac{1}{(kOH_{isoprene} - kOH_{MACR})} \times ln\left(1 - \frac{[a\text{-}MACR]}{[a\text{-}isoprene]} \times \frac{kOH_{MACR} - kOH_{isoprene}}{0.23 \times kOH_{isoprene}}\right) (Eq.\ 10)$$

$$[OH]\Delta t_{isoprene} = ([OH]\Delta t_{MVK} + [OH]\Delta t_{MACR})\ /\ 2 \quad (Eq.\ 11)$$

$$[e\text{-}isoprene] = [a\text{-}isoprene] \times exp(kOH_{isoprene}[OH]\Delta t_{isoprene}) \quad (Eq.\ 12)$$

*where kOH$_{isoprene}$, kOH$_{MVK}$, and kOH$_{MACR}$ are the reaction rate constants of isoprene (1 × 10⁻¹⁰ cm³ molecule⁻¹ s⁻¹), MVK (2.0 × 10⁻¹¹ cm³ molecule⁻¹ s⁻¹), and MACR (2.8 × 10⁻¹¹ cm³ molecule⁻¹ s⁻¹) with OH (Carter, 2010; Atkinson and Arey, 2003). [a-isoprene], [a-MVK], and [a-MACR] refer to the ambient concentrations of isoprene, MVK, and MACR, respectively.'*

**3.** Fig S4: The initial emission ratios between benzene and isoprene were obtained from Figure S4 for use in Eq. 1. However, the authors should explain why the fit slopes and their associated uncertainties do not appear to accurately represent the underlying data shown. For example, I would have expected a higher slope at Meishan than what is

presented. What is the R-squared for each of those fits? How much does the magnitude of the slope ultimately affect the calculation of the NO3 exposure?

Fig S5: Similar comment as above for Figure S4 .

**Response:**

In fact, the slope is not obtained by fitting all the data points. Since benzene is less reactive than styrene, the emitted ratio of benzene to styrene should be small compared to the ratio of their ambient concentrations. As the air mass is transported, the consumption of styrene is greater than that of benzene, thus increasing their ratios. Therefore, Similar to Figure S8 in (de Gouw et al., 2017), we select the lower ratios while excluding the lowest outliers to determine the emitted ratio. When the emitted ratios vary between $2.7 \pm 0.5$ ppbv ppbv$^{-1}$ at Meishan (Fig. S4), the corresponding changes in NO$_3$ exposure remain within 20%. When the emitted ratios of benzene to cis-2-butene vary between $6.5 \pm 1.0$ ppbv ppbv$^{-1}$ at Meishan (Fig. S5), the corresponding changes in O$_3$ exposure remain within 14%.

4. Line 298: I agree that there appears to be a diurnal variation of OFP at Chengdu in Figure S10, but there does not appear to be one in Deyang (particularly in Panel (a) of Figure S10) even though the text says there is one. Sunlight intensity is mentioned as a possible reason, but total OFP at night is similar to that of the day in Figure S10a. Could the authors explain what is happening here?

**Response:**

We delele the description about diurnal variations at Deyang. We revise the sentences (please check lines 345-349 in Results and discussion).

*'After considering the nighttime alkene consumption, the emitted alkene concentrations are close to those during the day at Deyang. Moreover, the MIR values of alkenes are generally higher across these VOCs chemical groups (Carter, 2010). These may explain why the total OFP at night is similar to that during the day at Deyang. The diurnal variations in the OFP at Chengdu are consistent with those of sunlight intensity.'*

5. Lines 312-314: Comparison is only made to the Song et al. (2018) paper, but Lines 39-41 suggest multiple studies reporting OFP for Chengdu and Deyang. I would like the authors to compare their results with those other cited studies as well in addition to the Song et al. paper.

**Response:**

We shortlist all reported top three VOCs species contributing to OFP in the Sichuan Basin, China in Table S3 (Wang et al., 2023; Kong et al., 2023; Chen et al., 2021; Xiong et al., 2021; Tan et al., 2020a; Tan et al., 2020b; Deng et al., 2019; Song et al., 2018). We revise the sentences (please check lines 360-365 in Results and discussion).

*'The top three species contributing to OFP among emitted VOCs are cis-2-butene, isoprene, m,p-xylene at Deyang; m,p-xylene, acetaldehyde, isoprene at Chengdu; and isoprene, ethylene, acetaldehyde at Meishan, respectively (Table 2). These results emphasize the importance of isoprene in $O_3$ formation. They differ from those based on ambient VOCs concentrations (Table 2) and those reported at Chengdu and Deyang from from 2016 to 2019, which are often within m,p-xylene, ethylene, toluene, and acetaldehyde (Table S3; Wang et al., 2023; Kong et al., 2023; Chen et al., 2021; Xiong et al., 2021; Tan et al., 2020a; Tan et al., 2020b; Deng et al., 2019; Song et al., 2018)'*

*Table S3 The reported top three VOCs species contributing to OFP based on ambient concentrations.*

| Name | Sampling time | VOCs Number | Type of sites | References |
|---|---|---|---|---|
| *Acetaldehyde, Ethylene, and m,p-Xylene; m,p-Xylene, Toluene, and Ethylene; Ethylene, m,p-Xylene, and Acetaldehyde; Acetaldehyde, m,p-Xylene, and Ethylene; Ethylene, Acetaldehyde, and m,p-Xylene; Ethylene, Acetaldehyde, and Toluene* | *May 2016 – January 2017* | *99* | *Urban Chengdu* | *(Tan et al., 2020b)* |
| *Ethylene, trans-2-Pentene, and Toluene* | *August 28, 2016 – October 7, 2016* | *94* | *Urban Chengdu* | *(Deng et al., 2019)* |
| *Ethylene, Propylene, and m,p-Xylene* | *October 27, 2016 – September 30, 2017* | *55* | *Urban Chengdu* | *(Song et al., 2018)* |
| *Propylene, 2-Butene, and 1-Butene; m,p-Xylene, Acetaldehyde, and Toluene; Acetaldehyde, m,p-Xylene, and o-Xylene* | *July 31, 2017 – August 31, 2017* | *99* | *Urban Chengdu* | *(Tan et al., 2020a)* |
| *m,p-Xylene, Toluene, and Ethylene; Ethylene, m,p-Xylene, and Toluene* | *June 1, 2018 – June 29, 2018* | *90* | *Urban Chengdu* | *(Xiong et al., 2021)* |
| *Ethylene, m,p-Xylene, and Toluene; m,p-Xylene, Toluene, and Ethylene; m,p-Xylene, Ethylene, and o-Xylene; Ethylene, m,p-Xylene, and Propylene* | *January 1, 2019 – December 31, 2019* | *56* | *Urban Chengdu* | *(Kong et al., 2023)* |
| *m,p-Xylene, Toluene, and o-Xylene* | *June to August 2019* | *122* | *Urban Chengdu* | *(Wang et al., 2023)* |
| *Acetaldehyde and Isoprene* | *August 20, 2019 – September 12, 2019* | *10* | *Rural Deyang* | *(Chen et al., 2021)* |

**6.** Comment on Abstract and Conclusion: The text for the abstract and conclusion are almost identical. I would suggest that the authors consider using these sections to reiterate how their modified method for calculating OFP is novel compared to past approaches. Is there a result that particularly stands out that the prior approach of simply using the observed/measured concentrations to calculate OFP missed?

**Response:**

We add details of the modified method and emphasize the importance of isoprene, which may often be overlooked in observed concentrations. We revise the sentences (please check lines 8-19 in Abstract and 369-383 in Conclusion).

*'To reduce the uncertainties in identifying the key volatile organic compounds (VOCs) species in ozone ($O_3$) formation from ambient VOCs concentrations, this study proposes a novel method to identify the key VOCs species within anthropogenic and biogenic emissions. The emitted VOCs concentrations are calculated during both night and day in summer using the nitrate radical, $O_3$, and hydroxyl radical reaction rates and ambient concentrations of 99 VOCs at Deyang, Chengdu, and Meishan, China. The emitted concentrations of alkenes and aromatics are higher than the ambient concentrations. The largest differences between emitted and ambient concentrations are 1.04 ppbv for cis-2-butene at Deyang, 0.81 ppbv for isoprene at Chengdu, and 1.79 ppbv for isoprene at Meishan, respectively. Due to secondary production, the emitted concentrations of oxygenated VOCs are lower than the ambient concentrations. The largest differences are -0.54 ppbv for acetone at Deyang, -0.58 ppbv for acetaldehyde at Chengdu, and -0.5 ppbv for acetone at Meishan, respectively. Based on the emitted concentrations, isoprene is one of the top three species contributing to $O_3$ formation at the three sites, which may be overlooked in observed concentrations. Comprehensively calculating the emitted VOCs concentrations enables the key VOCs species in $O_3$ formation to be accurately identified.'*

*'Using $NO_3$, $O_3$, and OH reaction rates and hourly ambient concentrations of 99 VOCs at Deyang, Chengdu, and Meishan, in Southwest China, the emitted VOCs concentrations are calculated during both night and day in summer. They are compared with the ambient concentrations in OFP. Currently, most studies identify the key VOCs species contributing to $O_3$ formation based on ambient VOC concentrations or by only considering the OH consumption of NMHCs. However, the emitted concentrations of VOCs, directly linked to MIR values, are more important for $O_3$ formation in the actual atmosphere than ambient VOCs concentrations. The average emitted concentrations of alkenes and aromatics are significantly higher than the ambient concentrations. The largest differences between emitted and ambient concentrations are 1.04 ppbv for cis-2-butene at Deyang, 0.81 ppbv for isoprene at Chengdu, and 1.79 ppbv for isoprene at Meishan, respectively. Because of the secondary production, the emitted OVOCs concentrations are lower than the ambient ones. The largest differences are -0.54 ppbv for acetone at Deyang, -0.58 ppbv for acetaldehyde at Chengdu, and -0.5 ppbv for acetone at Meishan, respectively. Based on the emitted VOCs concentrations, the top three species contributing to OFP are cis-2-butene, isoprene, and m,p-xylene at Deyang; m,p-xylene, acetaldehyde, and isoprene at Chengdu; and isoprene, ethylene, and acetaldehyde at Meishan, respectively. These results emphasize the importance of isoprene in $O_3$ formation and differ from those based on ambient concentrations. Comprehensively calculating the emitted concentrations of VOCs enables the accurate identification of the key VOCs*

7. In Section 2.2, refer to the upcoming sections (2.2.1, 2.2.2, etc.) as they are described in the text.

   Line 137: [m-alkenesj] is mentioned in the text, but Eq. 2 uses [a-alkenej]. Change one to be consistent with the other.

   Line 149: Same comment as Line 137. Use either a or m to be consistent.

   Figure 2 caption: Change wording of caption to denote that ambient VOC concentrations are colored black and initial VOC calculations are colored red. The (a) and (b) notation does not make sense.

   Line 262: Say OVOCs instead of oxygenated VOCs to be consistent.

   Line 290: Mention that this is for Meishan.

   Line 291: Should it say Figure S9ef?

   Line 298: A space is needed between 403.80 and μg m-3

   Line 309: Ioprene should be isoprene.

   Lines 315-318: consider moving this paragraph to the conclusion.

   **Response:** Done.

8. Check that appropriate verb tense is used throughout the manuscript. Numerous sentences were written in past tense when the present tense would be more appropriate.

   **Response:**

   We have paid for native English speakers to polish our writing. We have also reviewed the differences between original and polished manuscripts once again. The following paragraph is the explanation after being polished.

   *'The English in this document has been checked by at least two professional editors, both native speakers of English.'*

**References:**

Atkinson, R. and Arey, J.: Atmospheric degradation of volatile organic compounds, Chem. Rev., 103, 4605-4638, 10.1021/cr0206420, 2003.

Brown, S. S., Dubé, W. P., Fuchs, H., Ryerson, T. B., Wollny, A. G., Brock, C. A., Bahreini, R., Middlebrook, A. M., Neuman, J. A., Atlas, E., Roberts, J. M., Osthoff, H. D., Trainer, M., Fehsenfeld, F. C., and Ravishankara, A. R.: Reactive uptake coefficients for $N_2O_5$ determined from aircraft measurements during the Second Texas Air Quality Study: comparison to current model parameterizations, J. Geophys. Res.: Atmos., 114, 10.1029/2008jd011679, 2009a.

Brown, S. S., de Gouw, J. A., Warneke, C., Ryerson, T. B., Dubé, W. P., Atlas, E., Weber, R. J., Peltier, R. E., Neuman, J. A., Roberts, J. M., Swanson, A., Flocke, F., McKeen, S. A., Brioude, J., Sommariva, R., Trainer, M., Fehsenfeld, F. C., and Ravishankara, A. R.: Nocturnal isoprene oxidation over the Northeast United States in summer and its impact on reactive nitrogen partitioning and secondary organic aerosol, Atmos. Chem. Phys., 9, 3027-3042, 10.5194/acp-9-3027-2009, 2009b.

Carter, W. P. L.: Development of the SAPRC-07 chemical mechanism, Atmos. Environ., 44, 5324-5335, 10.1016/j.atmosenv.2010.01.026, 2010.

de Gouw, J. A., Gilman, J. B., Kim, S. W., Lerner, B. M., Isaacman-VanWertz, G., McDonald, B. C., Warneke, C., Kuster, W. C., Lefer, B. L., Griffith, S. M., Dusanter, S., Stevens, P. S., and Stutz, J.: Chemistry of volatile organic compounds in the Los Angeles Basin: nighttime removal of alkenes and determination of emission ratios, J. Geophys. Res.: Atmos., 122, 11, 843-811, 861, 10.1002/2017jd027459, 2017.

de Gouw, J. A., Middlebrook, A. M., Warneke, C., Goldan, P. D., Kuster, W. C., Roberts, J. M., Fehsenfeld, F. C., Worsnop, D. R., Canagaratna, M. R., Pszenny, A. A. P., Keene, W. C., Marchewka, M., Bertman, S. B., and Bates, T. S.: Budget of organic carbon in a polluted atmosphere: results from the New England air quality study in 2002, J. Geophys. Res.: Atmos., 110, D16305, 2005.

de Gouw, J. A., Gilman, J. B., Kim, S. W., Alvarez, S. L., Dusanter, S., Graus, M., Griffith, S. M., Isaacman-VanWertz, G., Kuster, W. C., Lefer, B. L., Lerner, B. M., McDonald, B. C., Rappenglück, B., Roberts, J. M., Stevens, P. S., Stutz, J., Thalman, R., Veres, P. R., Volkamer, R., Warneke, C., Washenfelder, R. A., and Young, C. J.: Chemistry of volatile organic compounds in the Los Angeles Basin: formation of oxygenated compounds and determination of emission ratios, J. Geophys. Res.: Atmos., 123, 2298-2319, 10.1002/2017jd027976, 2018.

Jacob, D. J. and Winner, D. A.: Effect of climate change on air quality, Atmos. Environ., 43, 51-63, 10.1016/j.atmosenv.2008.09.051, 2009.

Kong, L., Zhou, L., Chen, D. Y., Luo, L., Xiao, K., Chen, Y., Liu, H. F., Tan, Q. W., and Yang, F. M.: Atmospheric oxidation capacity and secondary pollutant formation potentials based on photochemical loss of VOCs in a megacity of the Sichuan Basin, China, Sci. Total Environ., 901, 166259, 10.1016/j.scitotenv.2023.166259, 2023.

Li, Z. J., He, L. Y., Ma, H. N., Peng, X., Tang, M. X., Du, K., and Huang, X. F.: Sources of atmospheric oxygenated volatile organic compounds in different air masses in Shenzhen, China, Environ Pollut, 122871, 10.1016/j.envpol.2023.122871, 2023.

Lyu, X. P., Li, H. Y., Lee, S. C., Xiong, E. Y., Guo, H., Wang, T., and de Gouw, J.: Significant biogenic source of oxygenated volatile organic compounds and the impacts on photochemistry at a regional background site in south China, Environ. Sci. Technol., 10.1021/acs.est.4c05656, 2024.

Ma, W., Feng, Z. M., Zhan, J. L., Liu, Y. C., Liu, P. F., Liu, C. T., Ma, Q. X., Yang, K., Wang, Y. F., He, H., Kulmala, M., Mu, Y. J., and Liu, J. F.: Influence of photochemical loss of volatile organic compounds on understanding ozone formation mechanism, Atmos. Chem. Phys., 22, 4841-4851, 10.5194/acp-22-4841-2022, 2022.

Paulot, F., Crounse, J. D., Kjaergaard, H. G., Kurten, A., St Clair, J. M., Seinfeld, J. H., and Wennberg, P. O.: Unexpected epoxide formation in the gas-phase photooxidation of isoprene, Science, 325, 730-733, 10.1126/science.1172910, 2009.

Roberts, J. M., Fehsenfeld, F. C., Liu, S. C., Bollinger, M. J., Hahn, C., Albritton, D. L., and Sievers, R. E.: Measurements of aromatic hydrocarbon ratios and $NO_x$ concentrations in the rural troposphere: observation of air mass photochemical aging and $NO_x$ removal, Atmos. Environ., 18, 2421-2432, 10.1016/0004-6981(84)90012-X, 1984.

Saunders, S. M., Jenkin, M. E., Derwent, R. G., and Pilling, M. J.: Protocol for the development of the Master Chemical Mechanism, MCM v3 (Part A): tropospheric degradation of non-aromatic volatile organic compounds, Atmos. Chem. Phys., 3, 161-180, 10.5194/acp-3-161-2003, 2003.

Shao, M., Wang, B., Lu, S. H., Yuan, B., and Wang, M.: Effects of Beijing Olympics control measures on reducing reactive hydrocarbon species, Environ. Sci. Technol., 45, 514-519, 2011.

Stroud, C. A., Roberts, J. M., Goldan, P. D., Kuster, W. C., Murphy, P. C., Williams, E. J., Hereid, D., Parrish, D., Sueper, D., Trainer, M., Fehsenfeld, F. C., Apel, E. C., Riemer, D., Wert, B., Henry, B., Fried, A., Martinez-Harder, M., Harder, H., Brune, W. H., Li, G., Xie, H., and Young, V. L.: Isoprene and its oxidation products, methacrolein and methyl vinyl ketone, at an urban forested site during the 1999 southern oxidants study, J. Geophys. Res.: Atmos., 106, 8035-8046, Doi 10.1029/2000jd900628, 2001.

Tan, Z. F., Lu, K. D., Jiang, M. Q., Su, R., Dong, H. B., Zeng, L. M., Xie, S. D., Tan, Q. W., and Zhang, Y. H.: Exploring ozone pollution in Chengdu, southwestern China: a case study from radical chemistry to $O_3$-VOC-$NO_x$ sensitivity, Sci. Total Environ., 636, 775-786, 10.1016/j.scitotenv.2018.04.286, 2018.

Wu, C. H., Wang, C. M., Wang, S. H., Wang, W. J., Yuan, B., Qi, J. P., Wang, B. L., Wang, H. L., Wang, C., Song, W., Wang, X. M., Hu, W. W., Lou, S. R., Ye, C. S., Peng, Y. W., Wang, Z. L., Huangfu, Y. B., Xie, Y., Zhu, M. N., Zheng, J. Y., Wang, X. M., Jiang, B., Zhang, Z. Y., and Shao, M.: Measurement report: important contributions of oxygenated compounds to emissions and chemistry of volatile organic compounds in urban air, Atmos. Chem. Phys., 20, 14769-14785, 10.5194/acp-20-14769-2020, 2020.

Wu, R. R. and Xie, S. D.: Spatial distribution of ozone formation in China derived from emissions of speciated volatile organic compounds, Environ. Sci. Technol., 51, 2574-2583, 10.1021/acs.est.6b03634, 2017.

Xiong, C., Wang, N., Zhou, L., Yang, F. M., Qiu, Y., Chen, J. H., Han, L., and Li, J. J.: Component characteristics and source apportionment of volatile organic compounds during summer and winter in downtown Chengdu, southwest China, Atmos. Environ., 258, 118485, 10.1016/j.atmosenv.2021.118485, 2021.

Yang, X. P., Lu, K. D., Ma, X. F., Liu, Y. H., Wang, H. C., Hu, R. Z., Li, X., Lou, S. R., Chen, S. Y., Dong, H. B., Wang, F. Y., Wang, Y. H., Zhang, G. X., Li, S. L., Yang, S. D., Yang, Y. M., Kuang, C. L., Tan, Z. F., Chen, X. R., Qiu, P. P., Zeng, L. M., Xie, P. H., and Zhang, Y. H.: Observations and modeling of OH and $HO_2$ radicals in Chengdu, China in summer 2019, Sci. Total Environ., 772, 144829, 10.1016/j.scitotenv.2020.144829, 2021.

Zheng, X. D., Ren, J., Hao, Y. F., and Xie, S. D.: Weekend-weekday variations, sources, and secondary transformation potential of volatile organic compounds in urban Zhengzhou, China, Atmos. Environ., 300, 119679, 10.1016/j.atmosenv.2023.119679, 2023.

Zou, Y., Guan, X. H., Flores, R. M., Yan, X. L., Fan, L. Y., Deng, T., Deng, X. J., and Ye, D. Q.: Revealing the influencing factors of an oxygenated volatile organic compounds (OVOCs) source apportionment model: a case study of a dense urban agglomeration in the winter, J. Geophys. Res.: Atmos., 129, 10.1029/2023jd039401, 2024.

---

## Author Comment (AC2)

**Authors' responses to review comments**

Atmospheric Chemistry and Physics (egusphere-2024-2568)

**Differences in the key volatile organic compound species between their emitted and ambient concentrations in ozone formation**

**Reviewer #2**

1. Zheng and Xie present a method for assessing the ozone forming potential of a series of volatile organic compounds measured in the Sichuan Basin, China. The method is based on observation of ambient concentrations of VOCs, followed by estimates of what is then referred to as "initial" VOCs, intended to represent the emitted amounts of these VOCs or the amount produced by secondary production in the atmosphere. Ozone formation potentials are then calculated using literature maximum incremental reactivities based on either observed or corrected VOCs.

The manuscript is clearly written and the figures are of good quality for presentation. It will be of interest to the readership of ACP.

**Response: Thank you for reviewing our manuscript.**

2. The major comment is that the classification of VOCs by chemical functional group rather than sources may lead to some errors. This is especially true for alkenes, but may also pertain to the oxygenates. Anthropogenic alkenes should be treated differently from biogenic alkenes (mainly isoprene), since these VOCs have very different sources. Division into these categories would make the analysis methods also self-consistent, although it may require some change in methodology for isoprene itself.

**Response:**

We have revised the overall classification method in Section 2.2. The emitted VOC concentrations are classified into three categories: anthropogenic sources in Section 2.2.1, biogenic sources in Section 2.2.2, and a combination of both in Section 2.2.3. In addition to secondary production, ten of the 13 OVOCs (except MTBE, MVK, and MACR) are also emitted from both anthropogenic and biogenic sources (Zou et al., 2024; Lyu et al., 2024; Li et al., 2023; Wu et al., 2020). We revise the sentences (please check lines 97-245 in Materials and methods). We also update the comparison between emitted and ambient VOC concentrations, as well as their corresponding OFP values in Results and discussion.

**2.2 Calculations of emitted VOCs concentrations**

Source classification is crucial for calculating emitted VOCs concentrations. NMHCs (except isoprene) and MTBE are generally emitted from anthropogenic activities. Isoprene is typically emitted from biogenic sources and oxidized into methyl vinyl ketone (MVK) and methacrolein (MACR). In addition to secondary production, ten of the 13 *OVOCs* (except MTBE, MVK, and MACR) are also emitted from both anthropogenic and biogenic sources (Zou et al., 2024; Lyu et al., 2024; Li et al., 2023; Wu et al., 2020). The emitted VOC concentrations are classified into three categories: anthropogenic sources in Section 2.2.1, biogenic sources in Section 2.2.2, and a combination of both in Section 2.2.3.

First, the major atmospheric oxidants for the consumption of emitted VOCs are NO3, *O3*, and *OH*. Due to the absence of sunlight, *OH* concentrations and photolysis rates are very low from 20:00 to 06:00 (Fig. S2). Either NO3 or  $O_3$  are the primary oxidants for the consumption of emitted alkene and styrene during the nighttime. During the local nighttime, emitted alkene and styrene concentrations are estimated through the NO3 or O3 exposure methods based on the relative loss rates of reported species between NO3 and O3 in the Los Angeles Basin (de Gouw et al., 2017). Other VOCs are excluded from the analysis due to their slow reaction rates with NO3 and O3 during nighttime. The NO3 or O3 exposure method indicates that the concentration ratios of a stable tracer species to a reactive tracer species would increase with both NO3 or O3 concentrations and reaction time after emissions. Emitted concentrations are calculated based on NO3 or O3 reaction rates and exposure. Unreported alkenes are classified through comparison with reported alkenes in reaction rates of both NO3 (kNO3) and O3 (kO3) (Fig. S3). For example, the nocturnal consumption of 1-butene is over 96% through reaction with O3 (de Gouw et al., 2017). The kO3 for 1-pentene is higher than the kO3 for 1-butene, but the kNO3 for 1pentene is lower than the kNO3 for 1-butene. Therefore, the emitted 1-pentene concentrations are estimated using the  $O_3$  exposure method. Briefly, styrene and 1,3butadiene are determined using the NO3 exposure method, while eight of the ten alkenes are determined using the O3 exposure method.

During the daytime from 7:00 to 19:00 (Fig. S2), NO3 is highly unstable and rapidly photolyzed. Therefore, VOCs consumption by its oxidation is negligible. Alkenes and styrene can react with both OH and O3. For alkenes and styrene, the ratio of the product of the OH reaction rates (Carter, 2010) and the ambient OH concentration in the Chengdu Plain ( $6.14 \times 10^6$  molecules cm-3; (Yang et al., 2021) to the product of the O3 reaction rates (Carter, 2010; Atkinson and Arey, 2003) and the ambient O3 concentration (45.71 ppbv) is 19.20. This indicates that OH predominantly consumes VOCs during the daytime. The emitted concentrations of NMHCs and MTBE are quantified during the daytime using the OH exposure method in Section 2.2.1. The OH exposure method is similar to the NO3 and O3 exposure methods,

Second, Brown et al. (2009b) calculated emitted isoprene concentrations during nighttime based on the steady-state NO3 production from the reaction of NO2 with O3 and its consumption by isoprene. The mean ratios of measured O3 to NO2 concentrations during nighttime are 4.64 ppbv ppbv-1 at Deyang, 1.42 ppbv ppbv-1 at Chengdu, and 2.23 ppbv ppbv-1 at Meishan. The reported method is not suitable for this study, because O3 concentrations must be much larger than the NO2 concentrations (Brown et al., 2009a).

Similar to styrene and 1,3-butadiene, emitted isoprene concentrations are determined using the NO3 exposure method during nighttime. During the day, emitted isoprene concentrations are calculated using the OH exposure method and the ambient concentrations of MVK and MACR. Emitted MVK and MACR concentrations are calculated based on their measured concentrations and isoprene consumption.

Third, emitted OVOCs concentrations are determined using the photochemical age method during the daytime, due to its primary emissions, secondary production, and consumption by both OH and photon (hv). Among meteorological factors, temperature is the primary driver of O3 production (Jacob and Winner, 2009), with the OH reaction rate showing small variations between 25°C and 35°C. For example, the reaction rate ratio for isoprene between these two temperatures is 0.96 (Saunders et al., 2003). Consequently, all VOCs reaction rate constants are adjusted for a temperature of 300 K.

2.2.1 NMHCs concentrations emitted by anthropogenic activities

For nighttime NO3 consumption, the emitted concentrations of styrene and 1,3butadiene are estimated using the NO3 exposure method (de Gouw et al., 2017):

$$[NO_3]\Delta t = \frac{1}{(kNO_{3benzene} - kNO_{3styrene})} \times \left[ ln \left( \frac{[e-benzene]}{[e-styrene]} \right) - ln \left( \frac{[a-benzene]}{[a-styrene]} \right) \right]$$
(Eq. 1)

$$[e-alkene_j] = [a-alkene_j] \times exp(kNO3_{alkene_j}[NO_3]\Delta t)$$
(Eq. 2)

where [NO3] and  $\Delta t$ , together referred to as NO3 exposure ([NO3] $\Delta t$ ), are the NO3 concentrations and nocturnal reaction time, respectively.  $kNO3_{benzene}$  and  $kNO3_{styrene}$  are the reaction rate constants of benzene ( $3.0 \times 10^{-17}$  cm3 molecule-1 s-1) and isoprene ( $1.5 \times 10^{-13}$  cm3 molecule-1 s-1) with NO3, respectively (Carter, 2010; Atkinson and Arey, 2003). [e-benzene]/[e-isoprene] is the emission ratio between benzene and isoprene. The estimated emission ratios are  $1.0 \pm 0.4$  ppbv ppbv-1 at Deyang,  $1.1 \pm 0.5$  ppbv ppbv-1 at Chengdu, and  $2.7 \pm 0.5$  ppbv ppbv-1 at Meishan based on the measured data with a low degree of nocturnal consumption, respectively (Fig. S4). [a-benzene]/[a-styrene] is the hourly ambient concentration ratio between benzene and isoprene. [a-alkenesj] and  $kNO3_{alkenej}$  refer to the ambient concentrations and NO3 reaction rate constants (Fig. S3) of species j in styrene or 1,3-butadiene, respectively.

For nighttime O3 consumption, the emitted concentrations of eight reactive alkenes are estimated using the O3 exposure method (de Gouw et al., 2017):

$$[O_3]\Delta t = \frac{1}{(kO_{3benzene} - kO_{3cis-2-butene})} \times \left[ ln \left( \frac{[e-benzene]}{[e-cis-2-butene]} \right) - ln \left( \frac{[a-benzene]}{[a-cis-2-butene]} \right) \right] \quad (Eq. 3)$$

$$[e-alkene_j] = [a-alkene_j] \times exp(kO3_{alkene_j}[O_3]\Delta t)$$
(Eq. 4)

where  $[O_3]$  and  $\Delta t$ , together referred to as  $O_3$  exposure  $([O_3]\Delta t)$ , are the  $O_3$  concentrations and nocturnal reaction time.  $kO_{3benzene}$  and  $kO_{3cis-2-butene}$  are the reaction rate constants of benzene  $(1.0 \times 10^{-20} \text{ cm}^3 \text{ molecule}^{-1} \text{ s}^{-1})$  and cis-2-butene  $(1.3 \times 10^{-16} \text{ cm}^3 \text{ molecule}^{-1} \text{ s}^{-1})$  with  $O_3$  (Carter, 2010; Atkinson and Arey, 2003). [e-benzene]/[e-cis-2-butene] is the emission ratios between benzene and cis-2-butene. The estimated emission

ratios are  $0.5 \pm 0.3ppbv ppbv^{-1}$  at Deyang,  $4.5 \pm 1.0ppbv ppbv^{-1}$  at Chengdu, and  $6.5 \pm 1.0ppbv ppbv^{-1}$  at Meishan based on measured data with a low degree of nocturnal consumption, respectively (Fig. S5). Similar to emission ratios of benzene to isoprene, emission ratios of benzene to cis-2-butene may remain consistent for each source. After mixing from different sources, the emission ratios obtained at different sampling sites may vary. [a-benzene]/[a-cis-2-butene] is the hourly ambient concentration ratio between benzene and cis-2-butene. [a-alkenesj] and  $kO3_{alkenej}$  refer to the ambient concentrations and O3 reaction rate constants (Fig. S3) of the species j in alkenes, respectively. Cis-2-butene is replaced with trans-2-butene at Chengdu due to the unavailability of cis-2-butene data.

For daytime OH consumption, the emitted concentrations of each NMHC [e-NMHC], including MTBE, are estimated using the OH exposure method (Ma et al., 2022; Shao et al., 2011; de Gouw et al., 2005; Roberts et al., 1984):

$$[OH] \Delta t = \frac{1}{(kOH_{ethylbenzene} - kOH_{m,p-xylenes})} \times \left[ ln \left( \frac{[e-ethylbenzene]}{[e-m,p-xylenes]} \right) - ln \left( \frac{[a-ethylbenzene]}{[a-m,p-xylenes]} \right) \right] (Eq. 5)$$

$$[e-NHMC_j] = [a-NHMC_j] \times exp(kOH_{NMHC_j}[OH] \Delta t)$$

$$(Eq. 6)$$

where [OH] and  $\Delta t$ , together referred to as OH exposure ([OH] $\Delta t$ ), are the OH concentrations and reaction time, respectively. [e-ethylbenzene]/[e-m,p-xylenes] is the emission ratio between ethylbenzene and m,p-xylenes (Fig. S6). The major source of ethylbenzene and m,p-xylenes in the Chengdu Plain is solvent use (Wu and Xie, 2017). There is a strong linear correlation between ethylbenzene and m,p-xylenes ( $R^2 = 0.96$ ).  $kOH_{ethylbenzene}$  and  $kOH_{m,p-xylenes}$  are the reaction rate constants of ethylbenzene ( $7.0 \times 10^{-12}$  cm3 molecule-1 s-1) and m,p-xylenes ( $1.9 \times 10^{-11}$  cm3 molecule-1 s-1) with OH, respectively (Carter, 2010). [a-NMHCj] and  $kOH_{NMHC_j}$  denote the hourly ambient concentrations and OH reaction rate constants (Fig. S7) of the species j in NMHCs, respectively.

2.2.2 Isoprene concentrations emitted by biogenic sources

For nighttime NO3 consumption, the emitted concentrations of isoprene are estimated using the NO3 exposure method (de Gouw et al., 2017):

$$[NO_3]\Delta t = \frac{1}{(kNO_{3MVK} - kNO_{3isoprene})} \times \left[ ln \left( \frac{[e-MVK]}{[e-isoprene]} \right) - ln \left( \frac{[a-MVK]}{[a-isoprene]} \right) \right]$$
(Eq. 7)

$$[e-isoprene] = [a-isoprene] \times exp(kNO3_{isoprene}[NO_3]\Delta t)$$

$$(Eq. 8)$$

where  $[NO_3]$  and  $\Delta t$ , together referred to as  $NO_3$  exposure  $([NO_3]\Delta t)$ , are the concentrations of  $NO_3$  and nocturnal reaction time, respectively.  $kNO_{3MVK}$  and  $kNO_{3isoprene}$  are the reaction rate constants of MVK ( $5.4 \times 10^{-18}$  cm3 molecule-1 s-1 with  $O_3$ ) and isoprene ( $6.8 \times 10^{-13}$  cm3 molecule-1 s-1) with  $NO_3$ , respectively (Carter, 2010; Atkinson and Arey, 2003). The  $kNO_{3MVK}$  value is very small. Due to the unavailability of the  $kNO_{3MVK}$  value,  $kO_{3MVK}$  is used as a substitute. [e-MVK]/[e-isoprene] is the emission ratio between MVK and isoprene. Although MVK and isoprene emissions are low at night, many field studies have demonstrated that they can accumulate in the early nighttime from

20:00 to 21:00 (Wennberg et al., 2018). Therefore, the measured MVK and isoprene concentrations in the early nighttime are the "emitted" concentrations for nighttime NO3 consumption. The estimated emission ratios are  $0.5 \pm 0.2$  ppbv ppbv-1 at Deyang,  $0.1 \pm$ 0.1ppbv ppbv-1 at Chengdu, and  $0.1 \pm 0.1$ ppbv ppbv-1 at Meishan from measured data with a low degree of nocturnal consumption, respectively (Fig. S8). There are no significant differences in the estimated emission ratios between early and late nighttime. Therefore, nighttime low MVk and isoprene emissions may not influence this calculation method. The emission ratios are directly linked to emission sources. After mixing from different sources, the emission ratios obtained at different sampling sites may vary. Although MVK and isoprene may originate from different sources during the nighttime, both anthropogenic and biogenic activities in the Chengdu Plain are relatively stable at nighttime based on both our unpublished results and the reported findings of a study using positive matrix factorization (Zheng et al., 2023; Kong et al., 2023; Xiong et al., 2021). Furthermore, as surrogates for traffic flows, the traffic congestion indices during the nighttime in Chengdu *remain relatively stable (https://jiaotong.baidu.com/congestion/city/urbanrealtime).* Therefore, their emission ratios may remain consistent. [a-MVK]/[a-isoprene] is the hourly ambient concentration ratio between MVK and isoprene. MACR is not used as a stable biogenic tracer due to its relatively high NO3 reaction rate  $(3.5 \times 10^{-15} \text{ cm}^3)$ molecule-1 s-1) compared to MVK.

For daytime OH consumption, the emitted isoprene concentrations are estimated using the OH exposure method in Eqs. (9-12) (Paulot et al., 2009; Stroud et al., 2001).

$$[OH] \Delta t_{MVK} = \frac{1}{(kOH_{isoprene} - kOH_{MVK})} \times ln \left( 1 - \frac{[a - MVK]}{[a - isoprene]} \times \frac{kOH_{MVK} - kOH_{isoprene}}{0.32 \times kOH_{isoprene}} \right) \quad (Eq. 9)$$

$$[OH] \Delta t_{MACR} = \frac{1}{(kOH_{isoprene} - kOH_{MACR})} \times ln \left( 1 - \frac{[a - MACR]}{[a - isoprene]} \times \frac{kOH_{MACR} - kOH_{isoprene}}{0.23 \times kOH_{isoprene}} \right) (Eq. 10)$$

$$[OH] \Delta t_{isoprene} = ([OH] \Delta t_{MVK} + [OH] \Delta t_{MACR}) / 2$$
(Eq. 11)

$$[e-isoprene] = [a-isoprene] \times exp(kOH_{isoprene}[OH] \Delta t_{isoprene})$$
(Eq. 12)

where  $kOH_{isoprene}$ ,  $kOH_{MVK}$ , and  $kOH_{MACR}$  are the reaction rate constants of isoprene  $(1 \times 10^{-10} \text{ cm}^3 \text{ molecule}^{-1} \text{ s}^{-1})$ , MVK  $(2.0 \times 10^{-11} \text{ cm}^3 \text{ molecule}^{-1} \text{ s}^{-1})$ , and MACR  $(2.8 \times 10^{-11} \text{ cm}^3 \text{ molecule}^{-1} \text{ s}^{-1})$ , with OH (Carter, 2010; Atkinson and Arey, 2003). [a-isoprene], [a-MVK], and [a-MACR] refer to the ambient concentrations of isoprene, MVK, and MACR, respectively.

For daytime OH and hv consumption, the emitted concentrations of MVK and MACR ([e-OVOC]) are estimated based on isoprene consumption:

[c-isoprene] = [e-isoprene] - [a-isoprene](Eq. 13)

$$[s-OVOC_j] = p \times [c-isoprene]$$
(Eq. 14)

 $[e-OVOC_{j}] = [a-OVOC_{j}] - [s-OVOC_{j}] + [c-isoprene] \times \frac{([a-OVOC_{j}] - [s-OVOC_{j}]) \times kOH^{*}_{OVOC_{j}}}{[a-isoprene] \times kOH_{isoprene}}$  (Eq. 15)

where [c-isoprene] indicates consumed concentrations of isoprene, which are equivalent to its emitted concentrations ([e-isoprene]) calculated from Eq. (12) minus the ambient concentrations ([a-isoprene]). [s-OVOCi] represents the secondary concentrations of species *j* in MVK or MACR produced from isoprene oxidation. The *p* values in Eq. (14) represent the molecular production from one molecular unit of isoprene consumption, with values of 0.32 for MVK and 0.23 for MACR, respectively (Paulot et al., 2009). Because the kOHMVK and kOHMACR values are 3.5 times and 5 times lower than those of kOHisoprene (Fig. S7), respectively, the ambient concentrations ([a-OVOCi]) are assumed to be instantaneous total concentrations in Eq. (15). Therefore,  $[e-OVOC_i]$  is approximately equal to  $[a-OVOC_i]$  minus  $[s-OVOC_i]$  plus the corresponding photochemical consumption, which is calculated using [c-isoprene] and the reaction rates. The photochemical consumption of OVOCs includes both photolysis and reaction with OH. Therefore, the total OVOCj loss rates (kOH\* OVOCj) are estimated based on the photolysis rate (Jovoci) and the loss rate with OH ([OH]kOHovoci). The ratio of the JNO2 to OH concentrations in the Sichuan Basin is similar to that reported in the Los Angeles Basin (Yang et al., 2021; de Gouw et al., 2018); therefore the Jovoci and [OH]kOHovoci may be comparable (Tan et al., 2018; de Gouw et al., 2018). Accordingly, we assume that the ratios (0.6 for MVK and MACR) of JOVOCi to [OH]kOHOVOCi established in the Los Angeles Basin (Fig. S9; (de Gouw et al., 2018) are applicable for estimating kOH\* OVOCj  $= (1 + 0.6) \times kOH_{OVOCi}$  in the Sichuan Basin.  $kOH_{isoprene}$  is the OH reaction rate constant of isoprene.

2.2.3 OVOCs concentrations emitted by both anthropogenic and biogenic sources

To differentiate secondary production and consumption, we estimate the emitted concentrations of ten of the 13 OVOCs ([e-OVOC]) during the daytime using the photochemical age method in Eqs. (16-17) (Wu et al., 2020; de Gouw et al., 2018; de Gouw et al., 2005). MTBE is excluded, with details provided in Section 2.2.1. The estimation methods for MVK and MACR are described in Section 2.2.2.

$$[a-OVOC_{j}] = ER_{OVOC_{j}} \times [a-benzene] \times exp(-(kOH_{OVOC_{j}}^{*} - kOH_{benzene})[OH]\Delta t) + ER_{HC} \times [a-benzene] \times \frac{kOH_{HC}}{kOH_{OVOC_{j}}^{*} - kOH_{HC}} \times \frac{exp(-kOH_{HC}[OH]\Delta t) - exp(-kOH_{OVOC_{j}}^{*}[OH]\Delta t)}{exp(-kOH_{benzene}[OH]\Delta t))} + ER_{HC} \times \frac{kOH_{HC}}{kOH_{OVOC_{j}}^{*} - kOH_{HC}} \times \frac{exp(-kOH_{HC}[OH]\Delta t) - exp(-kOH_{OVOC_{j}}^{*}[OH]\Delta t)}{exp(-kOH_{benzene}[OH]\Delta t))} + ER_{HC} \times \frac{kOH_{HC}}{kOH_{OVOC_{j}}^{*} - kOH_{HC}} \times \frac{exp(-kOH_{HC}[OH]\Delta t) - exp(-kOH_{OVOC_{j}}^{*}[OH]\Delta t)}{exp(-kOH_{benzene}[OH]\Delta t))} + ER_{HC} \times \frac{kOH_{HC}}{kOH_{OVOC_{j}}^{*} - kOH_{HC}} \times \frac{exp(-kOH_{HC}[OH]\Delta t) - exp(-kOH_{OVOC_{j}}^{*}[OH]\Delta t)}{exp(-kOH_{benzene}[OH]\Delta t)} + ER_{HC} \times \frac{exp(-kOH_{HC}[OH]\Delta t) - exp(-kOH_{OVOC_{j}}^{*}[OH]\Delta t)}{exp(-kOH_{benzene}[OH]\Delta t))} + ER_{HC} \times \frac{exp(-kOH_{HC}[OH]\Delta t) - exp(-kOH_{benzene}[OH]\Delta t)}{exp(-kOH_{benzene}[OH]\Delta t))} + ER_{HC} \times \frac{exp(-kOH_{HC}[OH]\Delta t) - exp(-kOH_{benzene}[OH]\Delta t)}{exp(-kOH_{benzene}[OH]\Delta t)} + ER_{HC} \times \frac{exp(-kOH_{HC}[OH]\Delta t) - exp(-kOH_{benzene}[OH]\Delta t)}{exp(-kOH_{benzene}[OH]\Delta t)} + ER_{HC} \times \frac{exp(-kOH_{benzene}[OH]\Delta t)}{exp(-kOH_{benzene}[OH$$

 $ER_{biogenic} \times [e\text{-}isoprene]$

$$[e-OVOC_j] = ER_{OVOC_j} \times [a-benzene] + ER_{biogenic} \times [e-isoprene]$$
(Eq. 17)

where the measured concentrations of species *j* in OVOCs ([*a*-OVOCj]) are equivalent to the sum of primary anthropogenic contributions, secondary anthropogenic contributions, and biogenic contributions, as represented sequentially in Eq. (16). Benzene is selected as the tracer of anthropogenic primary sources due to the dominance of combustion and industrial VOCs emissions in the Sichuan Basin (Wu and Xie, 2017) and its relatively low OH reaction rate. ERovocj and ERHC are the emission ratios of species *j* in OVOCs and hydrocarbons to benzene, respectively. We assume that the ratios (*R*) for Jovocj to [OH]kOHovocj established in the Los Angeles Basin (Fig. S9; (de Gouw et al., 2018) are applicable for estimating  $kOH^* OVOCj = (1 + R) \times kOHovocj$  in the Sichuan Basin. OH exposure is calculated using Eq. (5).  $ER_{biogenic}$  is the emission ratio between OVOCs and isoprene from biogenic sources. [e-isoprene] is estimated by Eqs. (9-12). The ERovocj,  $ER_{HC}$ ,  $kOH_{HC}$ , and  $ER_{biogenic}$  values are determined using the nonlinear least-squares fit.'

3. English language comment: The paper uses past tense frequently where present tense make more sense for English writing. If possible, suggest proofreading by a native English speaker. As one example, the first line of the manuscript (line 24) is better write as "Volatile organic compounds are key species in ozone formation".

**Response:**

We have paid for native English speakers to polish our writing. We have also reviewed the differences between original and polished manuscripts once again. The following paragraph is the explanation after being polished.

'The English in this document has been checked by at least two professional editors, both native speakers of English.'

**4.** Line 28: The terminology of "initial VOCs" to indicate VOC emissions is somewhat confusing. Would suggest a different label for these compounds to make this clear, as in "emitted VOCs".

**Response:**

The term "initial VOCs" has been replaced with "emitted VOCs (e-VOCs)." Please review the text, figures, and tables accordingly.

5. Line 81: replace "moving" with "removing" and "particulate matters" with "particulate matter"

**Response:** Done (please check line 83 in Materials and Methods).

**6.** Line 97-98: The "NO3 or O3 exposure method" is not defined. How is this used to estimate emissions?

**Response:**

We define the NO3 or O3 exposure method and explain the theoretical basis for estimating emitted concentrations (please check lines 109-111 in Materials and Methods). The details of calculation methods are given in Eqs. 1-4 and 7-8.

'The NO3 or O3 exposure method indicates that the concentration ratios of a stable tracer species to a reactive tracer species would increase with both NO3 or O3 concentrations and reaction time after emissions.'

7. Line 117 and equation 1: Benzene and isoprene are not co-emitted species, so no relationship or emission ratio would be expected between these species. Furthermore, these species have very different diel emissions profiles, with isoprene not emitted at night but strongly peaked during daytime. The indicated dependence of the benzene to isoprene ratio is therefore not likely to be applicable to this pair. A different approach would be required to relate the concentrations of these species that lack a common emission source. Other pairs, such as benzene – styrene, or benzene – butadiene that have common anthropogenic sources should be amenable to this method.

**Response:**

We choose benzene and styrene for the calculation of NMHCs concentrations emitted by anthropogenic activities. We revise the sentences (please check lines 138-149 in Materials and Methods).

'For nighttime NO3 consumption, the emitted concentrations of styrene and 1,3-butadiene are estimated using the NO3 exposure method (de Gouw et al., 2017):

$$[NO_3] \Delta t = \frac{1}{(kNO_{benzene} - kNO_{styrene})} \times \left[ ln \left( \frac{[e-benzene]}{[e-styrene]} \right) - ln \left( \frac{[a-benzene]}{[a-styrene]} \right) \right]$$
(Eq. 1)

$$[e-alkene_j] = [a-alkene_j] \times exp(kNO3_{alkene_j}[NO_3]\Delta t)$$
(Eq. 2)

where [NO3] and  $\Delta t$ , together referred to as NO3 exposure ([NO3] $\Delta t$ ), are the NO3 concentrations and nocturnal reaction time, respectively.  $kNO3_{benzene}$  and  $kNO3_{styrene}$  are the reaction rate constants of benzene ( $3.0 \times 10^{-17}$  cm3 molecule-1 s-1) and isoprene ( $1.5 \times 10^{-13}$  cm3 molecule-1 s-1) with NO3, respectively (Carter, 2010; Atkinson and Arey, 2003). [e-benzene]/[e-isoprene] is the emission ratio between benzene and isoprene. The estimated emission ratios are  $1.0 \pm 0.4$  ppbv ppbv-1 at Deyang,  $1.1 \pm 0.5$  ppbv ppbv-1 at Chengdu, and  $2.7 \pm 0.8$  ppbv ppbv-1 at Meishan based on the measured data with a low degree of nocturnal consumption, respectively (Fig. S4). [a-benzene]/[a-styrene] is the hourly ambient concentration ratio between benzene and isoprene. [a-alkenesj] and  $kNO3_{alkenej}$  refer to the ambient concentrations and NO3 reaction rate constants (Fig. S3) of species j in styrene or 1,3-butadiene, respectively.'

**8.** Line 271 and figure S10: It does not necessarily make sense to calculate an OFP during nighttime hours.

**Response:**

We compare potential differences in the OFP between ambient and emitted VOCs concentrations even during the nighttime. If not suitable, we also emphasize the distinction between daytime and nighttime in Figures S10 and S11. We revise the sentences (please check lines 316-350 in Results and discussion).

'Although the OFP is primarily associated with daytime  $O_3$  production, nighttime OFP is also calculated to compare the potential differences in the contributions of VOCs and emitted VOCs to the OFP (Figures 10 and 11). The temporal variation trends in the

OFP values are relatively consistent at the three sites based on emitted and ambient TVOCs concentrations (Figure S10ab). The OFP based on emitted and ambient TVOCs concentrations ranges from 65.60  $\mu$ g m-3 to 1476.28  $\mu$ g m-3 and from 55.86  $\mu$ g m-3 to 827.30  $\mu$ g m-3, respectively, at Deyang. Among VOCs chemical groups, alkenes exhibit the greatest variations in OFP, with ranges of 15.12  $\mu$ g m-3 to 1081.41  $\mu$ g m-3 and 7.87  $\mu$ g m-3 to 282.78  $\mu$ g m-3, respectively. The highest OFP based on the emitted TVOCs concentrations was found at 16:00 on 11 August 2019, primarily due to the 1081.41  $\mu$ g m-3 contribution from alkenes. At this time, the OFP based on the ambient TVOCs concentrations is only 327.41  $\mu$ g m-3. The ratio of the OFP between the emitted and ambient TVOCs concentrations exceeds 4.5. This discrepancy is mainly because of the large consumption of alkenes at that time. If the consumption of alkenes is not considered, their contributions to the OFP would be greatly underestimated.

The OFP based on emitted and ambient TVOCs concentrations ranges from 47.47  $\mu$ g m-3 to 1143.74  $\mu$ g m-3 and from 39.21  $\mu$ g m-3 to 819.61  $\mu$ g m-3, respectively at Chengdu (Figure S10cd). Alkenes exhibit the greatest variations in OFP among VOCs chemical groups, with ranges from 12.34  $\mu$ g m-3 to 780.85  $\mu$ g m-3 and from 4.94  $\mu$ g m-3 to 336.58  $\mu$ g m-3, respectively. The highest OFP based on the emitted TVOCs concentrations was observed at 12:00 on 30 August 2019, which is primarily due to the contributions of 500.64  $\mu$ g m-3 from alkenes and 494.89  $\mu$ g m-3 from aromatics. At this time, the OFP based on the ambient TVOCs concentrations is 819.61  $\mu$ g m-3. The OFP based on the emitted and ambient TVOCs concentrations are similar, indicating that photochemical consumption is relatively low at Chengdu compared to Deyang. Furthermore, the OFP of the ambient OVOCs concentrations is 171.94  $\mu$ g m-3, which is higher than the OFP of emitted concentrations at 88.40  $\mu$ g m-3. Compared to the other two sites, emissions from solvent use are higher around Chengdu. This leads to higher emissions of aromatics, which significantly contributes to the OFP.

The OFP based on emitted and ambient TVOCs concentrations ranges from 59.30  $\mu g m^{-3}$  to 1351.58  $\mu g m^{-3}$  and from 48.43  $\mu g m^{-3}$  to 1077.27  $\mu g m^{-3}$ , respectively at Meishan (Figure S10ef). Alkenes display the greatest variations in OFP, ranging from 13.98  $\mu g m^{-3}$  to 1133.53  $\mu g m^{-3}$  and from 7.62  $\mu g m^{-3}$  to 864.52  $\mu g m^{-3}$ , respectively. The highest OFP of the emitted TVOCs concentrations was found at 18:00 on 17 August 2019, due to the major contribution of 1133.53  $\mu g m^{-3}$  from alkenes. Because of low photochemical consumption and secondary OVOCs formation, the difference in the OFP between emitted and ambient TVOCs concentrations was relatively low, at about 25% at this time. Compared to the other two sites, there are more biogenic isoprene emissions around Meishan, which contributes to the OFP of alkenes.

The highest hourly OFP values of emitted TVOCs concentrations are  $391.07 \ \mu g \ m^{-3}$  at 12:00 at Deyang,  $432.12 \ \mu g \ m^{-3}$  at 12:00 at Chengdu, and  $403.80 \ \mu g \ m^{-3}$  at 18:00 at Meishan, respectively (Figure S11). After considering the nighttime alkene consumption, the emitted alkene concentrations are close to those during the day at Deyang. Moreover,

the MIR values of alkenes are generally higher across these VOCs chemical groups (Carter, 2010). These may explain why the total OFP at night is similar to that during the day at Deyang. The diurnal variations in the OFP at Chengdu are consistent with those of sunlight intensity. The OFP at Meishan is primarily driven by isoprene. The surrounding bamboo forest acts as a source of biogenic emissions. The accumulation of isoprene causes the highest concentrations, and thus OFP, occurring at 18:00.'

9. Line 309: Isoprene rather than Ioprene

**Response:** Done (please check line 358 in Results and discussion).

**Reviewer #3**

1. Having hourly concentrations of 99 VOCs and OVOCs at three different field sites in the Sichuan Basin, China, Zheng and Xie calculate the ozone formation potential (OFP) for these species using the maximum incremental reactivity (MIR) method. However, unlike previous studies that have calculated OFPs using the observed/measured concentration of the VOC or OVOC, the authors correct these concentrations to estimate what the concentrations of these species would have been before being consumed or produced in the atmosphere (something they refer to as "initial" concentrations). The authors show that using the "initial" concentration of a species can impact the relative importance of that species in forming ozone.

The paper would be appropriate for inclusion in ACP after the following comments are addressed:

**Response: Thank you for reviewing our manuscript.**

2. Eqs. 1, 3, and 5, there should be a brief explanation for each on why benzene and isoprene, benzene and cis-2-butene, and ethylbenzene and m,p-xylenes are paired to calculate the NO3, O3, and OH exposures, respectively. For example, in the NO3 exposure, benzene and isoprene are paired, but they have completely different emission sources and do not seem well correlated in Figure S4, so explaining why they are paired would be beneficial to explain for the reader.

**Response:**

We have revised the overall classification method in Section 2.2. The emitted VOC concentrations are classified into three categories: anthropogenic sources in Section 2.2.1, biogenic sources in Section 2.2.2, and a combination of both in Section 2.2.3. VOCs species pairs with common sources were selected to calculate the NO3, O3, and OH exposure in Sections 2.2.1 and 2.2.2. We revise the sentences (please check lines 97-131 and 138-207 in Materials and methods). We also update the comparison between emitted

and ambient VOC concentrations, as well as their corresponding OFP values in Results and discussion.

**2.2** Calculations of emitted VOCs concentrations**

Source classification is crucial for calculating emitted VOCs concentrations. NMHCs (except isoprene) and MTBE are generally emitted from anthropogenic activities. Isoprene is typically emitted from biogenic sources and oxidized into methyl vinyl ketone (MVK) and methacrolein (MACR). In addition to secondary production, ten of the 13 OVOCs (except MTBE, MVK, and MACR) are also emitted from both anthropogenic and biogenic sources (Zou et al., 2024; Lyu et al., 2024; Li et al., 2023; Wu et al., 2020). The emitted VOC concentrations are classified into three categories: anthropogenic sources in Section 2.2.1, biogenic sources in Section 2.2.2, and a combination of both in Section 2.2.3.

First, the major atmospheric oxidants for the consumption of emitted VOCs are NO3, *O3*, and *OH*. Due to the absence of sunlight, *OH* concentrations and photolysis rates are very low from 20:00 to 06:00 (Fig. S2). Either NO3 or O3 are the primary oxidants for the consumption of emitted alkene and styrene during the nighttime. During the local nighttime, emitted alkene and styrene concentrations are estimated through the  $NO_3$  or  $O_3$ exposure methods based on the relative loss rates of reported species between  $NO_3$  and  $O_3$ in the Los Angeles Basin (de Gouw et al., 2017). Other VOCs are excluded from the analysis due to their slow reaction rates with  $NO_3$  and  $O_3$  during nighttime. The  $NO_3$  or  $O_3$  exposure method indicates that the concentration ratios of a stable tracer species to a reactive tracer species would increase with both NO3 or O3 concentrations and reaction time after emissions. Emitted concentrations are calculated based on NO3 or O3 reaction rates and exposure. Unreported alkenes are classified through comparison with reported alkenes in reaction rates of both NO3 (kNO3) and O3 (kO3) (Fig. S3). For example, the nocturnal consumption of 1-butene is over 96% through reaction with  $O_3$  (de Gouw et al., 2017). The kO3 for 1-pentene is higher than the kO3 for 1-butene, but the kNO3 for 1pentene is lower than the kNO3 for 1-butene. Therefore, the emitted 1-pentene concentrations are estimated using the  $O_3$  exposure method. Briefly, styrene and 1,3butadiene are determined using the NO3 exposure method, while eight of the ten alkenes are determined using the O3 exposure method.

During the daytime from 7:00 to 19:00 (Fig. S2), NO3 is highly unstable and rapidly photolyzed. Therefore, VOCs consumption by its oxidation is negligible. Alkenes and styrene can react with both OH and O3. For alkenes and styrene, the ratio of the product of the OH reaction rates (Carter, 2010) and the ambient OH concentration in the Chengdu Plain ( $6.14 \times 10^6$  molecules cm-3; (Yang et al., 2021) to the product of the O3 reaction rates (Carter, 2010; Atkinson and Arey, 2003) and the ambient O3 concentration (45.71 ppbv) is 19.20. This indicates that OH predominantly consumes VOCs during the daytime. The emitted concentrations of NMHCs and MTBE are quantified during the daytime using

the OH exposure method in Section 2.2.1. The OH exposure method is similar to the NO3 and O3 exposure methods,

Second, Brown et al. (2009b) calculated emitted isoprene concentrations during nighttime based on the steady-state NO3 production from the reaction of NO2 with O3 and its consumption by isoprene. The mean ratios of measured O3 to NO2 concentrations during nighttime are 4.64 ppbv ppbv-1 at Deyang, 1.42 ppbv ppbv-1 at Chengdu, and 2.23 ppbv ppbv-1 at Meishan. The reported method is not suitable for this study, because O3 concentrations must be much larger than the NO2 concentrations (Brown et al., 2009a). Similar to styrene and 1,3-butadiene, emitted isoprene concentrations are determined using the NO3 exposure method during nighttime. During the day, emitted isoprene concentrations are calculated using the OH exposure method and the ambient concentrations of MVK and MACR. Emitted MVK and MACR concentrations are calculated based on their measured concentrations and isoprene consumption.'

2.2.1 NMHCs concentrations emitted by anthropogenic activities

For nighttime NO3 consumption, the emitted concentrations of styrene and 1,3butadiene are estimated using the NO3 exposure method (de Gouw et al., 2017):

$$[NO_3]\Delta t = \frac{1}{(kNO_{3benzene} - kNO_{3styrene})} \times \left[ ln \left( \frac{[e-benzene]}{[e-styrene]} \right) - ln \left( \frac{[a-benzene]}{[a-styrene]} \right) \right]$$
(Eq. 1)

$$[e-alkene_j] = [a-alkene_j] \times exp(kNO3_{alkene_j}[NO_3]\Delta t)$$
(Eq. 2)

where  $[NO_3]$  and  $\Delta t$ , together referred to as NO3 exposure  $([NO_3]\Delta t)$ , are the NO3 concentrations and nocturnal reaction time, respectively.  $kNO_{3benzene}$  and  $kNO_{3styrene}$  are the reaction rate constants of benzene  $(3.0 \times 10^{-17} \text{ cm}^3 \text{ molecule}^{-1} \text{ s}^{-1})$  and isoprene  $(1.5 \times 10^{-13} \text{ cm}^3 \text{ molecule}^{-1} \text{ s}^{-1})$  with NO3, respectively (Carter, 2010; Atkinson and Arey, 2003). [e-benzene]/[e-isoprene] is the emission ratio between benzene and isoprene. The estimated emission ratios are  $1.0 \pm 0.4 \text{ ppbv} \text{ ppbv}^{-1}$  at Deyang,  $1.1 \pm 0.5 \text{ ppbv} \text{ ppbv}^{-1}$  at Chengdu, and  $2.7 \pm 0.5 \text{ ppbv} \text{ ppbv}^{-1}$  at Meishan based on the measured data with a low degree of nocturnal consumption, respectively (Fig. S4). [a-benzene]/[a-styrene] is the hourly ambient concentration ratio between benzene and isoprene. [a-alkenesj] and  $kNO_{3alkenej}$  refer to the ambient concentrations and NO3 reaction rate constants (Fig. S3) of species j in styrene or 1,3-butadiene, respectively.

For nighttime  $O_3$  consumption, the emitted concentrations of eight reactive alkenes are estimated using the  $O_3$  exposure method (de Gouw et al., 2017):

$$[O_3] \Delta t = \frac{1}{(kO_{3benzene} - kO_{3cis-2-butene})} \times \left[ ln \left( \frac{[e-benzene]}{[e-cis-2-butene]} \right) - ln \left( \frac{[a-benzene]}{[a-cis-2-butene]} \right) \right] \quad (Eq. 3)$$

$$[e-alkene_j] = [a-alkene_j] \times exp(kO3_{alkene_j}[O_3]\Delta t)$$
(Eq. 4)

where  $[O_3]$  and  $\Delta t$ , together referred to as  $O_3$  exposure  $([O_3]\Delta t)$ , are the  $O_3$  concentrations and nocturnal reaction time.  $kO_{3benzene}$  and  $kO_{3cis-2-butene}$  are the reaction rate constants of benzene  $(1.0 \times 10^{-20} \text{ cm}^3 \text{ molecule}^{-1} \text{ s}^{-1})$  and cis-2-butene  $(1.3 \times 10^{-16} \text{ cm}^3 \text{ molecule}^{-1} \text{ s}^{-1})$  with  $O_3$  (Carter, 2010; Atkinson and Arey, 2003). [e-benzene]/[e-cis-2-

butene] is the emission ratios between benzene and cis-2-butene. The estimated emission ratios are  $0.5 \pm 0.3ppbv ppbv^{-1}$  at Deyang,  $4.5 \pm 1.0ppbv ppbv^{-1}$  at Chengdu, and  $6.5 \pm 1.0ppbv ppbv^{-1}$  at Meishan based on measured data with a low degree of nocturnal consumption, respectively (Fig. S5). Similar to emission ratios of benzene to isoprene, emission ratios of benzene to cis-2-butene may remain consistent for each source. After mixing from different sources, the emission ratios obtained at different sampling sites may vary. [a-benzene]/[a-cis-2-butene] is the hourly ambient concentration ratio between benzene and cis-2-butene. [a-alkenesj] and  $kO3_{alkenej}$  refer to the ambient concentrations and  $O_3$  reaction rate constants (Fig. S3) of the species j in alkenes, respectively. Cis-2butene is replaced with trans-2-butene at Chengdu due to the unavailability of cis-2-butene data.

For daytime OH consumption, the emitted concentrations of each NMHC [e-NMHC], including MTBE, are estimated using the OH exposure method (Ma et al., 2022; Shao et al., 2011; de Gouw et al., 2005; Roberts et al., 1984):

$$[OH] \Delta t = \frac{1}{(kOH_{ethylbenzene} - kOH_{m,p-xylenes})} \times \left[ ln \left( \frac{[e-ethylbenzene]}{[e-m,p-xylenes]} \right) - ln \left( \frac{[a-ethylbenzene]}{[a-m,p-xylenes]} \right) \right] (Eq. 5)$$

$$[e-NHMC_j] = [a-NHMC_j] \times exp(kOH_{NMHC_j}[OH] \Delta t)$$

$$(Eq. 6)$$

where [OH] and  $\Delta t$ , together referred to as OH exposure ([OH] $\Delta t$ ), are the OH concentrations and reaction time, respectively. [e-ethylbenzene]/[e-m,p-xylenes] is the emission ratio between ethylbenzene and m,p-xylenes (Fig. S6). The major source of ethylbenzene and m,p-xylenes in the Chengdu Plain is solvent use (Wu and Xie, 2017). There is a strong linear correlation between ethylbenzene and m,p-xylenes ( $R^2 = 0.96$ ).  $kOH_{ethylbenzene}$  and  $kOH_{m,p-xylenes}$  are the reaction rate constants of ethylbenzene ( $7.0 \times 10^{-12} \text{ cm}^3 \text{ molecule}^{-1} \text{ s}^{-1}$ ) and m,p-xylenes ( $1.9 \times 10^{-11} \text{ cm}^3 \text{ molecule}^{-1} \text{ s}^{-1}$ ) with OH, respectively (Carter, 2010). [a-NMHCj] and  $kOH_{NMHC_j}$  denote the hourly ambient concentrations and OH reaction rate constants (Fig. S7) of the species j in NMHCs, respectively.

2.2.2 Isoprene concentrations emitted by biogenic sources

For nighttime NO3 consumption, the emitted concentrations of isoprene are estimated using the NO3 exposure method (de Gouw et al., 2017):

$$[NO_3]\Delta t = \frac{1}{(kNO_{3MVK} - kNO_{3isoprene})} \times \left[ ln \left( \frac{[e-MVK]}{[e-isoprene]} \right) - ln \left( \frac{[a-MVK]}{[a-isoprene]} \right) \right]$$
(Eq. 7)

$$[e-isoprene] = [a-isoprene] \times exp(kNO3_{isoprene}[NO_3]\Delta t)$$
(Eq. 8)

where [NO3] and  $\Delta t$ , together referred to as NO3 exposure ([NO3] $\Delta t$ ), are the concentrations of NO3 and nocturnal reaction time, respectively.  $kNO3_{MVK}$  and  $kNO3_{isoprene}$  are the reaction rate constants of MVK ( $5.4 \times 10^{-18} \text{ cm}^3 \text{ molecule}^{-1} \text{ s}^{-1}$  with O3) and isoprene ( $6.8 \times 10^{-13} \text{ cm}^3 \text{ molecule}^{-1} \text{ s}^{-1}$ ) with NO3, respectively (Carter, 2010; Atkinson and Arey, 2003). The  $kNO3_{MVK}$  value is very small. Due to the unavailability of the  $kNO3_{MVK}$  value,  $kO3_{MVK}$  is used as a substitute. [e-MVK]/[e-isoprene] is the emission ratio between MVK and isoprene. Although MVK and isoprene emissions are low at night,

many field studies have demonstrated that they can accumulate in the early nighttime from 20:00 to 21:00 (Wennberg et al., 2018). Therefore, the measured MVK and isoprene concentrations in the early nighttime are the "emitted" concentrations for nighttime NO3 consumption. The estimated emission ratios are  $0.5 \pm 0.2$  ppbv ppbv-1 at Devang,  $0.1 \pm$ 0.1ppbv ppbv-1 at Chengdu, and  $0.1 \pm 0.1$ ppbv ppbv-1 at Meishan from measured data with a low degree of nocturnal consumption, respectively (Fig. S8). There are no significant differences in the estimated emission ratios between early and late nighttime. Therefore, nighttime low MVk and isoprene emissions may not influence this calculation method. The emission ratios are directly linked to emission sources. After mixing from different sources, the emission ratios obtained at different sampling sites may vary. Although MVK and isoprene may originate from different sources during the nighttime, both anthropogenic and biogenic activities in the Chengdu Plain are relatively stable at nighttime based on both our unpublished results and the reported findings of a study using positive matrix factorization (Zheng et al., 2023; Kong et al., 2023; Xiong et al., 2021). Furthermore, as surrogates for traffic flows, the traffic congestion indices during the nighttime in Chengdu *remain relatively stable (https://jiaotong.baidu.com/congestion/city/urbanrealtime).* Therefore, their emission ratios may remain consistent. [a-MVK]/[a-isoprene] is the hourly ambient concentration ratio between MVK and isoprene. MACR is not used as a stable biogenic tracer due to its relatively high NO3 reaction rate  $(3.5 \times 10^{-15} \text{ cm}^3)$ molecule-1 s-1) compared to MVK.

For daytime OH consumption, the emitted isoprene concentrations are estimated using the OH exposure method in Eqs. (9-12) (Paulot et al., 2009; Stroud et al., 2001).

$$[OH] \Delta t_{MVK} = \frac{1}{(kOH_{isoprene} - kOH_{MVK})} \times ln \left( 1 - \frac{[a - MVK]}{[a - isoprene]} \times \frac{kOH_{MVK} - kOH_{isoprene}}{0.32 \times kOH_{isoprene}} \right) \quad (Eq. 9)$$

$$[OH] \Delta t_{MACR} = \frac{1}{(kOH_{isoprene} - kOH_{MACR})} \times ln \left(1 - \frac{[a - MACR]}{[a - isoprene]} \times \frac{kOH_{MACR} - kOH_{isoprene}}{0.23 \times kOH_{isoprene}}\right) (Eq. 10)$$

$$[OH] \Delta t_{isoprene} = ([OH] \Delta t_{MVK} + [OH] \Delta t_{MACR}) / 2$$
(Eq. 11)

$$[e-isoprene] = [a-isoprene] \times exp(kOH_{isoprene}[OH] \Delta t_{isoprene})$$
(Eq. 12)

where  $kOH_{isoprene}$ ,  $kOH_{MVK}$ , and  $kOH_{MACR}$  are the reaction rate constants of isoprene  $(1 \times 10^{-10} \text{ cm}^3 \text{ molecule}^{-1} \text{ s}^{-1})$ , MVK  $(2.0 \times 10^{-11} \text{ cm}^3 \text{ molecule}^{-1} \text{ s}^{-1})$ , and MACR  $(2.8 \times 10^{-11} \text{ cm}^3 \text{ molecule}^{-1} \text{ s}^{-1})$  with OH (Carter, 2010; Atkinson and Arey, 2003). [a-isoprene], [a-MVK], and [a-MACR] refer to the ambient concentrations of isoprene, MVK, and MACR, respectively.'

**3.** Fig S4: The initial emission ratios between benzene and isoprene were obtained from Figure S4 for use in Eq. 1. However, the authors should explain why the fit slopes and their associated uncertainties do not appear to accurately represent the underlying data shown. For example, I would have expected a higher slope at Meishan than what is

presented. What is the R-squared for each of those fits? How much does the magnitude of the slope ultimately affect the calculation of the NO3 exposure?

Fig S5: Similar comment as above for Figure S4.

**Response:**

In fact, the slope is not obtained by fitting all the data points. Since benzene is less reactive than styrene, the emitted ratio of benzene to styrene should be small compared to the ratio of their ambient concentrations. As the air mass is transported, the consumption of styrene is greater than that of benzene, thus increasing their ratios. Therefore, Similar to Figure S8 in (de Gouw et al., 2017), we select the lower ratios while excluding the lowest outliers to determine the emitted ratio. When the emitted ratios vary between  $2.7 \pm 0.5$  ppbv ppbv-1 at Meishan (Fig. S4), the corresponding changes in NO3 exposure remain within 20%. When the emitted ratios of benzene to cis-2-butene vary between  $6.5 \pm 1.0$  ppbv ppbv-1 at Meishan (Fig. S5), the corresponding changes in O3 exposure remain within 14%.

4. Line 298: I agree that there appears to be a diurnal variation of OFP at Chengdu in Figure S10, but there does not appear to be one in Deyang (particularly in Panel (a) of Figure S10) even though the text says there is one. Sunlight intensity is mentioned as a possible reason, but total OFP at night is similar to that of the day in Figure S10a. Could the authors explain what is happening here?

**Response:**

We delele the description about diurnal variations at Deyang. We revise the sentences (please check lines 345-349 in Results and discussion).

'After considering the nighttime alkene consumption, the emitted alkene concentrations are close to those during the day at Deyang. Moreover, the MIR values of alkenes are generally higher across these VOCs chemical groups (Carter, 2010). These may explain why the total OFP at night is similar to that during the day at Deyang. The diurnal variations in the OFP at Chengdu are consistent with those of sunlight intensity.'

5. Lines 312-314: Comparison is only made to the Song et al. (2018) paper, but Lines 39-41 suggest multiple studies reporting OFP for Chengdu and Deyang. I would like the authors to compare their results with those other cited studies as well in addition to the Song et al. paper.

**Response:**

We shortlist all reported top three VOCs species contributing to OFP in the Sichuan Basin, China in Table S3 (Wang et al., 2023; Kong et al., 2023; Chen et al., 2021; Xiong et al., 2021; Tan et al., 2020a; Tan et al., 2020b; Deng et al., 2019; Song et al., 2018). We revise the sentences (please check lines 360-365 in Results and discussion). 'The top three species contributing to OFP among emitted VOCs are cis-2-butene, isoprene, m,p-xylene at Deyang; m,p-xylene, acetaldehyde, isoprene at Chengdu; and isoprene, ethylene, acetaldehyde at Meishan, respectively (Table 2). These results emphasize the importance of isoprene in O3 formation. They differ from those based on ambient VOCs concentrations (Table 2) and those reported at Chengdu and Deyang from from 2016 to 2019, which are often within m,p-xylene, ethylene, toluene, and acetaldehyde (Table S3; Wang et al., 2023; Kong et al., 2023; Chen et al., 2021; Xiong et al., 2021; Tan et al., 2020a; Tan et al., 2020b; Deng et al., 2019; Song et al., 2018)'

Table S3 The reported top three VOCs species contributing to OFP based on ambient concentrations.

| Name                                                                                                                                                                                                                                                  | Sampling time                            | VOCs
Number | Type of sites | References           |
|-------------------------------------------------------------------------------------------------------------------------------------------------------------------------------------------------------------------------------------------------------|------------------------------------------|----------------|---------------|----------------------|
| Acetaldehyde, Ethylene, and m,p-Xylene;
m,p-Xylene, Toluene, and Ethylene;
Ethylene, m,p-Xylene, and Acetaldehyde;
Acetaldehyde, m,p-Xylene, and Ethylene;
Ethylene, Acetaldehyde, and m,p-Xylene;
Ethylene, Acetaldehyde, and Toluene | May 2016–
January 2017                | 99             | Urban Chengdu | (Tan et al., 2020b)  |
| Ethylene, trans-2-Pentene, and Toluene                                                                                                                                                                                                                | August 28, 2016 –
October 7, 2016     | 94             | Urban Chengdu | (Deng et al., 2019)  |
| Ethylene, Propylene, and m,p-Xylene                                                                                                                                                                                                                   | October 27, 2016 –
September 30, 2017 | 55             | Urban Chengdu | (Song et al., 2018)  |
| Propylene, 2-Butene, and 1-Butene;
m,p-Xylene, Acetaldehyde, and Toluene;
Acetaldehyde, m,p-Xylene, and o-Xylene                                                                                                                                | July 31, 2017 –
August 31, 2017       | 99             | Urban Chengdu | (Tan et al., 2020a)  |
| m,p-Xylene, Toluene, and Ethylene;
Ethylene, m,p-Xylene, and Toluene                                                                                                                                                                               | June 1, 2018 –
June 29, 2018          | 90             | Urban Chengdu | (Xiong et al., 2021) |
| Ethylene, m,p-Xylene, and Toluene;
m,p-Xylene, Toluene, and Ethylene;
m,p-Xylene, Ethylene, and o-Xylene;
Ethylene, m,p-Xylene, and Propylene                                                                                                | January 1, 2019 –
December 31, 2019   | 56             | Urban Chengdu | (Kong et al., 2023)  |
| m,p-Xylene, Toluene, and o-Xylene                                                                                                                                                                                                                     | June to August 2019                      | 122            | Urban Chengdu | (Wang et al., 2023)  |
| Acetaldehyde and Isoprene                                                                                                                                                                                                                             | August 20, 2019 –
September 12, 2019  | 10             | Rural Deyang  | (Chen et al., 2021)  |

6. Comment on Abstract and Conclusion: The text for the abstract and conclusion are almost identical. I would suggest that the authors consider using these sections to reiterate how their modified method for calculating OFP is novel compared to past approaches. Is there a result that particularly stands out that the prior approach of simply using the observed/measured concentrations to calculate OFP missed?

**Response:**

We add details of the modified method and emphasize the importance of isoprene, which may often be overlooked in observed concentrations. We revise the sentences (please check lines 8-19 in Abstract and 369-383 in Conclusion).

'To reduce the uncertainties in identifying the key volatile organic compounds (VOCs) species in ozone (O3) formation from ambient VOCs concentrations, this study proposes a novel method to identify the key VOCs species within anthropogenic and biogenic emissions. The emitted VOCs concentrations are calculated during both night and day in summer using the nitrate radical, O3, and hydroxyl radical reaction rates and ambient concentrations of 99 VOCs at Deyang, Chengdu, and Meishan, China. The emitted concentrations of alkenes and aromatics are higher than the ambient concentrations. The largest differences between emitted and ambient concentrations are 1.04 ppbv for cis-2-butene at Devang, 0.81 ppbv for isoprene at Chengdu, and 1.79 ppbv for isoprene at Meishan, respectively. Due to secondary production, the emitted concentrations of oxygenated VOCs are lower than the ambient concentrations. The largest differences are -0.54 ppbv for acetone at Deyang, -0.58 ppbv for acetaldehyde at Chengdu, and -0.5 ppbv for acetone at Meishan, respectively. Based on the emitted concentrations, isoprene is one of the top three species contributing to O3 formation at the three sites, which may be overlooked in observed concentrations. Comprehensively calculating the emitted VOCs concentrations enables the key VOCs species in O3 formation to be accurately identified.'

'Using NO3, O3, and OH reaction rates and hourly ambient concentrations of 99 VOCs at Devang, Chengdu, and Meishan, in Southwest China, the emitted VOCs concentrations are calculated during both night and day in summer. They are compared with the ambient concentrations in OFP. Currently, most studies identify the key VOCs species contributing to O3 formation based on ambient VOC concentrations or by only considering the OH consumption of NMHCs. However, the emitted concentrations of VOCs, directly linked to MIR values, are more important for O3 formation in the actual atmosphere than ambient VOCs concentrations. The average emitted concentrations of alkenes and aromatics are significantly higher than the ambient concentrations. The largest differences between emitted and ambient concentrations are 1.04 ppbv for cis-2butene at Devang, 0.81 ppbv for isoprene at Chengdu, and 1.79 ppbv for isoprene at Meishan, respectively. Because of the secondary production, the emitted OVOCs concentrations are lower than the ambient ones. The largest differences are -0.54 ppbv for acetone at Deyang, -0.58 ppbv for acetaldehyde at Chengdu, and -0.5 ppbv for acetone at Meishan, respectively. Based on the emitted VOCs concentrations, the top three species contributing to OFP are cis-2-butene, isoprene, and m,p-xylene at Deyang; m,p-xylene, acetaldehyde, and isoprene at Chengdu; and isoprene, ethylene, and acetaldehyde at Meishan, respectively. These results emphasize the importance of isoprene in O3 formation and differ from those based on ambient concentrations. Comprehensively calculating the emitted concentrations of VOCs enables the accurate identification of the key VOCs

**species contributing to the OFP.'**

7. In Section 2.2, refer to the upcoming sections (2.2.1, 2.2.2, etc.) as they are described in the text.

Line 137: [m-alkenesj] is mentioned in the text, but Eq. 2 uses [a-alkenej]. Change one to be consistent with the other.

Line 149: Same comment as Line 137. Use either a or m to be consistent.

Figure 2 caption: Change wording of caption to denote that ambient VOC concentrations are colored black and initial VOC calculations are colored red. The (a) and (b) notation does not make sense.

Line 262: Say OVOCs instead of oxygenated VOCs to be consistent.

Line 290: Mention that this is for Meishan.

Line 291: Should it say Figure S9ef?

Line 298: A space is needed between 403.80 and µg m-3

Line 309: Ioprene should be isoprene.

Lines 315-318: consider moving this paragraph to the conclusion.

**Response: Done.**

**8.** Check that appropriate verb tense is used throughout the manuscript. Numerous sentences were written in past tense when the present tense would be more appropriate.

**Response:**

We have paid for native English speakers to polish our writing. We have also reviewed the differences between original and polished manuscripts once again. The following paragraph is the explanation after being polished.

'The English in this document has been checked by at least two professional editors, both native speakers of English.'

**References:**

Atkinson, R. and Arey, J.: Atmospheric degradation of volatile organic compounds, Chem. Rev., 103, 4605-4638, 10.1021/cr0206420, 2003.

Brown, S. S., Dubé, W. P., Fuchs, H., Ryerson, T. B., Wollny, A. G., Brock, C. A., Bahreini, R., Middlebrook, A. M., Neuman, J. A., Atlas, E., Roberts, J. M., Osthoff, H. D., Trainer, M., Fehsenfeld, F. C., and Ravishankara, A. R.: Reactive uptake coefficients for N2O5 determined from aircraft measurements during the Second Texas Air Quality Study: comparison to current model parameterizations, J. Geophys. Res.: Atmos., 114, 10.1029/2008jd011679, 2009a.

Brown, S. S., de Gouw, J. A., Warneke, C., Ryerson, T. B., Dubé, W. P., Atlas, E., Weber, R. J., Peltier, R. E., Neuman, J. A., Roberts, J. M., Swanson, A., Flocke, F., McKeen, S. A., Brioude, J., Sommariva, R., Trainer, M., Fehsenfeld, F. C., and Ravishankara, A. R.: Nocturnal isoprene oxidation over the Northeast United States in summer and its impact on reactive nitrogen partitioning and secondary organic aerosol, Atmos. Chem. Phys., 9, 3027-3042, 10.5194/acp-9-3027-2009, 2009b.

Carter, W. P. L.: Development of the SAPRC-07 chemical mechanism, Atmos. Environ., 44, 5324-5335, 10.1016/j.atmosenv.2010.01.026, 2010.

de Gouw, J. A., Gilman, J. B., Kim, S. W., Lerner, B. M., Isaacman-VanWertz, G., McDonald, B. C., Warneke, C., Kuster, W. C., Lefer, B. L., Griffith, S. M., Dusanter, S., Stevens, P. S., and Stutz, J.: Chemistry of volatile organic compounds in the Los Angeles Basin: nighttime removal of alkenes and determination of emission ratios, J. Geophys. Res.: Atmos., 122, 11, 843-811, 861, 10.1002/2017jd027459, 2017.

de Gouw, J. A., Middlebrook, A. M., Warneke, C., Goldan, P. D., Kuster, W. C., Roberts, J. M., Fehsenfeld, F. C., Worsnop, D. R., Canagaratna, M. R., Pszenny, A. A. P., Keene, W. C., Marchewka, M., Bertman, S. B., and Bates, T. S.: Budget of organic carbon in a polluted atmosphere: results from the New England air quality study in 2002, J. Geophys. Res.: Atmos., 110, D16305, 2005.

de Gouw, J. A., Gilman, J. B., Kim, S. W., Alvarez, S. L., Dusanter, S., Graus, M., Griffith, S. M., Isaacman-VanWertz, G., Kuster, W. C., Lefer, B. L., Lerner, B. M., McDonald, B. C., Rappenglück, B., Roberts, J. M., Stevens, P. S., Stutz, J., Thalman, R., Veres, P. R., Volkamer, R., Warneke, C., Washenfelder, R. A., and Young, C. J.: Chemistry of volatile organic compounds in the Los Angeles Basin: formation of oxygenated compounds and determination of emission ratios, J. Geophys. Res.: Atmos., 123, 2298-2319, 10.1002/2017jd027976, 2018.

Jacob, D. J. and Winner, D. A.: Effect of climate change on air quality, Atmos. Environ., 43, 51-63, 10.1016/j.atmosenv.2008.09.051, 2009.

Kong, L., Zhou, L., Chen, D. Y., Luo, L., Xiao, K., Chen, Y., Liu, H. F., Tan, Q. W., and Yang, F. M.: Atmospheric oxidation capacity and secondary pollutant formation potentials based on photochemical loss of VOCs in a megacity of the Sichuan Basin, China, Sci. Total Environ., 901, 166259, 10.1016/j.scitotenv.2023.166259, 2023.

Li, Z. J., He, L. Y., Ma, H. N., Peng, X., Tang, M. X., Du, K., and Huang, X. F.: Sources of atmospheric oxygenated volatile organic compounds in different air masses in Shenzhen, China, Environ Pollut, 122871, 10.1016/j.envpol.2023.122871, 2023.

Lyu, X. P., Li, H. Y., Lee, S. C., Xiong, E. Y., Guo, H., Wang, T., and de Gouw, J.: Significant biogenic source of oxygenated volatile organic compounds and the impacts on photochemistry at a regional background site in south China, Environ. Sci. Technol., 10.1021/acs.est.4c05656, 2024.

Ma, W., Feng, Z. M., Zhan, J. L., Liu, Y. C., Liu, P. F., Liu, C. T., Ma, Q. X., Yang, K., Wang, Y. F., He, H., Kulmala, M., Mu, Y. J., and Liu, J. F.: Influence of photochemical loss of volatile organic compounds on understanding ozone formation mechanism, Atmos. Chem. Phys., 22, 4841-4851, 10.5194/acp-22-4841-2022, 2022.

Paulot, F., Crounse, J. D., Kjaergaard, H. G., Kurten, A., St Clair, J. M., Seinfeld, J. H., and Wennberg, P. O.: Unexpected epoxide formation in the gas-phase photooxidation of isoprene, Science, 325, 730-733, 10.1126/science.1172910, 2009.

Roberts, J. M., Fehsenfeld, F. C., Liu, S. C., Bollinger, M. J., Hahn, C., Albritton, D. L., and Sievers, R. E.: Measurements of aromatic hydrocarbon ratios and  $NO_x$  concentrations in the rural troposphere: observation of air mass photochemical aging and  $NO_x$  removal, Atmos. Environ., 18, 2421-2432, 10.1016/0004-6981(84)90012-X, 1984.

Saunders, S. M., Jenkin, M. E., Derwent, R. G., and Pilling, M. J.: Protocol for the development of the Master Chemical Mechanism, MCM v3 (Part A): tropospheric degradation of non-aromatic volatile organic compounds, Atmos. Chem. Phys., 3, 161-180, 10.5194/acp-3-161-2003, 2003.

Shao, M., Wang, B., Lu, S. H., Yuan, B., and Wang, M.: Effects of Beijing Olympics control measures on reducing reactive hydrocarbon species, Environ. Sci. Technol., 45, 514-519, 2011.

Stroud, C. A., Roberts, J. M., Goldan, P. D., Kuster, W. C., Murphy, P. C., Williams, E. J., Hereid, D., Parrish, D., Sueper, D., Trainer, M., Fehsenfeld, F. C., Apel, E. C., Riemer, D., Wert, B., Henry, B., Fried, A., Martinez-Harder, M., Harder, H., Brune, W. H., Li, G., Xie, H., and Young, V. L.: Isoprene and its oxidation products, methacrolein and methyl vinyl ketone, at an urban forested site during the 1999 southern oxidants study, J. Geophys. Res.: Atmos., 106, 8035-8046, Doi 10.1029/2000jd900628, 2001.

Tan, Z. F., Lu, K. D., Jiang, M. Q., Su, R., Dong, H. B., Zeng, L. M., Xie, S. D., Tan, Q. W., and Zhang, Y. H.: Exploring ozone pollution in Chengdu, southwestern China: a case study from radical chemistry to  $O_3$ -VOC-NOx sensitivity, Sci. Total Environ., 636, 775-786, 10.1016/j.scitotenv.2018.04.286, 2018.

Wu, C. H., Wang, C. M., Wang, S. H., Wang, W. J., Yuan, B., Qi, J. P., Wang, B. L., Wang, H. L., Wang, C., Song, W., Wang, X. M., Hu, W. W., Lou, S. R., Ye, C. S., Peng, Y. W., Wang, Z. L., Huangfu, Y. B., Xie, Y., Zhu, M. N., Zheng, J. Y., Wang, X. M., Jiang, B., Zhang, Z. Y., and Shao, M.: Measurement report: important contributions of oxygenated compounds to emissions and chemistry of volatile organic compounds in urban air, Atmos. Chem. Phys., 20, 14769-14785, 10.5194/acp-20-14769-2020, 2020.

Wu, R. R. and Xie, S. D.: Spatial distribution of ozone formation in China derived from emissions of speciated volatile organic compounds, Environ. Sci. Technol., 51, 2574-2583, 10.1021/acs.est.6b03634, 2017.

Xiong, C., Wang, N., Zhou, L., Yang, F. M., Qiu, Y., Chen, J. H., Han, L., and Li, J. J.: Component characteristics and source apportionment of volatile organic compounds during summer and winter in downtown Chengdu, southwest China, Atmos. Environ., 258, 118485, 10.1016/j.atmosenv.2021.118485, 2021.

Yang, X. P., Lu, K. D., Ma, X. F., Liu, Y. H., Wang, H. C., Hu, R. Z., Li, X., Lou, S. R., Chen,
S. Y., Dong, H. B., Wang, F. Y., Wang, Y. H., Zhang, G. X., Li, S. L., Yang, S. D., Yang, Y.
M., Kuang, C. L., Tan, Z. F., Chen, X. R., Qiu, P. P., Zeng, L. M., Xie, P. H., and Zhang, Y.
H.: Observations and modeling of OH and HO2 radicals in Chengdu, China in summer 2019,
Sci. Total Environ., 772, 144829, 10.1016/j.scitotenv.2020.144829, 2021.

Zheng, X. D., Ren, J., Hao, Y. F., and Xie, S. D.: Weekend-weekday variations, sources, and secondary transformation potential of volatile organic compounds in urban Zhengzhou, China, Atmos. Environ., 300, 119679, 10.1016/j.atmosenv.2023.119679, 2023.

Zou, Y., Guan, X. H., Flores, R. M., Yan, X. L., Fan, L. Y., Deng, T., Deng, X. J., and Ye, D. Q.: Revealing the influencing factors of an oxygenated volatile organic compounds (OVOCs) source apportionment model: a case study of a dense urban agglomeration in the winter, J. Geophys. Res.: Atmos., 129, 10.1029/2023jd039401, 2024.

---

## Author Response (AR2)

**Authors' responses to review comments**

Atmospheric Chemistry and Physics (egusphere-2024-2568)

**Differences in the key volatile organic compound species between their emitted and ambient concentrations in ozone formation**

1. The authors have responded to the comments of the reviewers well. There are a couple of edits necessary prior to publication:

**Response:** Thank you for reviewing our manuscript.

2. Line 133: change "consumption by both OH and photon" to "losses either by photolysis or photooxidation"

   Line 144: should be e-benzene/e-styrene?

   Line 209: change "For daytime OH and hv consumption" to "For photolysis and photooxidation loss"

   Fig. 2: please move legend out of panel a (make it at top or something that is easier to find) with it having larger font and/or bold. Right now it takes a while to find, which makes interpreting the figure more difficult.

   Please ensure references to supplemental figures include an S. Right now, there are references to figures that I assume are in main document (no s) but they don't exist or match text.

   Line 351: Change "According to Table 2" to "As shown in Table 2"

**Response:** Done.

3. Please reflect biogenic, anthropogenic, and OVOCs in figure as there is not a direct connection now between those methods and the figures. The figures with them by class can remain in SI for reference.

**Response:**

The emitted and ambient concentrations of OVOCs are presented in Figure 2, Tables S1 and S2. Emitted concentrations can be categorized into anthropogenic and biogenic sources. Anthropogenic emissions account for 26%±21% to 62%±12% of the total emitted concentrations, while biogenic sources contribute 38%±12% to 74%±21%. Details will be referenced to Zheng et al., unpublished, Characteristics and sources of volatile organic compounds and their influences on ozone and fine particulate nitrate production sensitivities at five sites in the Chengdu Plain, China.

4. Fig 1 & 3: please consider a different color for either alkenes or alkanes to better differentiate them for color blind people

**Response:** Done. We also revised the Figures S7, S10, and S11.

5. Please review abstract and conclusion with the guidelines prior to resubmission: https://www.atmospheric-chemistry-and-hysics.net/policies/guidelines_for_authors.html

**Response:**

We revise the sentences (please check lines 8-21 in Abstract and 371-384 in Conclusion).

*'Volatile organic compounds (VOCs) emissions and their secondary transformations play a significant role in ozone ($O_3$) formation. Previous studies have often relied on ambient VOCs concentrations to identify key VOCs species. However, ambient concentrations represent the residual concentrations after the emitted VOCs have been consumed, which can introduce substantial uncertainties. To address this issue, this study proposes a novel method to identify the key VOCs species in both anthropogenic and biogenic emissions. The emitted VOCs concentrations are calculated during both nighttime and daytime in summer using the nitrate radical, $O_3$, and hydroxyl radical reaction rates and ambient concentrations of 99 VOCs measured at Deyang, Chengdu, and Meishan, China. The emitted concentrations of alkenes and aromatics are higher than the ambient concentrations. The largest differences between emitted and ambient concentrations are 1.04 ppbv for cis-2-butene at Deyang, 0.81 ppbv for isoprene at Chengdu, and 1.79 ppbv for isoprene at Meishan. In contrast, due to secondary production, the emitted concentrations of oxygenated VOCs are lower than the ambient concentrations. The largest differences are -0.54 ppbv for acetone at Deyang, -0.58 ppbv for acetaldehyde at Chengdu, and -0.5 ppbv for acetone at Meishan. Based on the emitted concentrations, isoprene is one of the top three species contributing to $O_3$ formation at all three sites, which may be overlooked in observed concentrations. Comprehensive calculation of the emitted VOCs concentrations enables the key VOCs species in $O_3$ formation to be accurately identified.'*

*'Using $NO_3$, $O_3$, and OH reaction rates along with hourly ambient concentrations of 99 VOCs measured at Deyang, Chengdu, and Meishan in Southwest China, we calculate the emitted VOCs concentrations during both nighttime and daytime in summer. These emitted VOCs concentrations are compared with the ambient concentrations in terms of their OFP. Because the emitted VOCs concentrations are directly linked to MIR values, this novel method substantially enhances the accurate identification of the key VOCs species in $O_3$ formation. The emitted concentrations of alkenes and aromatics are significantly higher than the ambient concentrations. In contrast, because of the secondary production, the emitted OVOCs concentrations are lower than the ambient ones. Based on the emitted VOCs concentrations, the top three species contributing to OFP are cis-2-*

*butene, isoprene, and m,p-xylene at Deyang; m,p-xylene, acetaldehyde, and isoprene at Chengdu; and isoprene, ethylene, and acetaldehyde at Meishan. These results emphasize the importance of isoprene in $O_3$ formation and differ from those based on ambient concentrations. While many current environmental policies focus on reducing emissions of non-isoprene alkenes and aromatics, our study shows that isoprene emissions are also important. Therefore, the control of isoprene emissions should be considered in the mitigation of $O_3$ pollution. Our study provides new insights into improving the scientific understanding of the VOCs emissions, their secondary transformations, and serves as a reference for managing key VOCs species in future control strategies.'*